# In vivo single-cell lineage tracing in zebrafish using high-resolution infrared laser-mediated gene induction microscopy

Sicong He[1,2,3†], Ye Tian[2,3,4†], Shachuan Feng[2,3,4†], Yi Wu[2,3,4], Xinwei Shen[5], Kani Chen[5], Yingzhu He[1,2,3], Qiqi Sun[1,2,3], Xuesong Li[1,2,3], Jin Xu[6*], Zilong Wen[2,3,4*], Jianan Y Qu[1,2,3*]

[1]Department of Electronic and Computer Engineering, The Hong Kong University of Science and Technology, Kowloon, China; [2]State Key Laboratory of Molecular Neuroscience, The Hong Kong University of Science and Technology, Kowloon, China; [3]Center of Systems Biology and Human Health, The Hong Kong University of Science and Technology, Kowloon, China; [4]Division of Life Science, The Hong Kong University of Science and Technology, Kowloon, China; [5]Department of Mathematics, The Hong Kong University of Science and Technology, Kowloon, China; [6]Division of Cell, Developmental and Integrative Biology, School of Medicine, South China University of Technology, Guangzhou, China

*For correspondence:
xujin@scut.edu.cn (JX);
zilong@ust.hk (ZW);
eequ@ust.hk (JYQ)

†These authors contributed equally to this work

Competing interests: The authors declare that no competing interests exist.

**Abstract** Heterogeneity broadly exists in various cell types both during development and at homeostasis. Investigating heterogeneity is crucial for comprehensively understanding the complexity of ontogeny, dynamics, and function of specific cell types. Traditional bulk-labeling techniques are incompetent to dissect heterogeneity within cell population, while the new single-cell lineage tracing methodologies invented in the last decade can hardly achieve high-fidelity single-cell labeling and long-term in-vivo observation simultaneously. In this work, we developed a high-precision infrared laser-evoked gene operator heat-shock system, which uses laser-induced CreER[T2] combined with loxP-DsRedx-loxP-GFP reporter to achieve precise single-cell labeling and tracing. In vivo study indicated that this system can precisely label single cell in brain, muscle and hematopoietic system in zebrafish embryo. Using this system, we traced the hematopoietic potential of hemogenic endothelium (HE) in the posterior blood island (PBI) of zebrafish embryo and found that HEs in the PBI are heterogeneous, which contains at least myeloid unipotent and myeloid-lymphoid bipotent subtypes.

## Introduction

Since new cells are generated from pre-existing cells, the frequently asked questions are what progenies are generated from certain pre-existing cells and how they contribute to the organism (*Child, 1906*; *Hoppe et al., 2014*; *Spanjaard and Junker, 2017*). To address these questions, fate mapping has been widely used as a crucial methodology to identify progenies of the targeted cells and to trace their location, differentiation and functional dynamics (*Kretzschmar and Watt, 2012*). Hematopoiesis, the process of forming blood cells (*Dharampuriya et al., 2017*), is an outstanding paradigm for studying these issues in different animal models (*Höfer et al., 2016*; *Jagannathan-Bogdan and Zon, 2013*; *Orkin and Zon, 2008*).

Zebrafish has natural advantages on hematopoietic fate mapping owing to the external development, transparent embryo body and its highly conserved hematopoiesis (*Jagannathan-Bogdan and Zon, 2013*; *Stachura and Traver, 2016*). Taking these advantages, permanent genetic marking, photo-convertible labeling and in vivo time-lapse imaging have been employed to monitor the

**eLife digest** Animals begin life as a single cell that then divides to become a complex organism with many different types of cells. Every time a cell divides, each of its two daughter cells can either stay the same type as their parent or adopt a different identity. Once a cell acquires an identity, it usually cannot 'go back' and choose another. Eventually, this process will produce daughter cells with the identity of a specific tissue or organ and that cannot divide further.

Multipotent cells are cells that can produce daughter cells with different identities, including other multipotent cells. These cells can usually give rise to different cell types in a specific organ, and generate more cells to replace any cells that die in that organ. Tracking the cells descended from a multipotent cell in a specific tissue can provide information about how the tissue develops.

Hemogenic endothelium cells produce the multipotent cells that give rise to two types of white blood cells: myeloid cells and lymphoid cells. Myeloid cells include innate immune cells that protect the body from infection non-specifically; while lymphoid cells include T cells and B cells with receptors that detect specific bacteria or viruses. It remains unclear whether each of these two cell types originate from a single population of hemogenic endothelium cells or from two distinct subpopulations.

He et al. have now developed a new optical technique to label a single hemogenic endothelium cell in a zebrafish and track the cell and its descendants. This method revealed that there are at least two distinct populations of hemogenic endothelium cells. One of them can give rise to both lymphoid and myeloid cells, while the other can only give rise to myeloid cells.

These findings shed light on the mechanisms of blood formation, and potentially could provide useful tools to study the development of diseases such as leukemia. Additionally, the single-cell labeling technology He et al. have developed could be applied to study the development of other tissues and organs.

generation, mobilization and lineage specification of hematopoietic stem/progenitor cells (HSPCs) in zebrafish (*Murayama et al., 2006*; *Jin et al., 2007*; *Bertrand et al., 2010*; *Kissa and Herbomel, 2010*). Recently, the infrared laser-evoked gene operator (IR-LEGO) microscope heating system (*Deguchi, 2009*; *Kamei et al., 2009*) has been demonstrated as a powerful tool for bulk cell tracing with high temporal-spatial resolution (*Shimada et al., 2013*; *Okuyama et al., 2013*; *Xu et al., 2015*; *Tian et al., 2017*; *Singhal and Shaham, 2017*; *He et al., 2018*; *Henninger et al., 2017*). In this system, an infrared (IR) laser is used to generate local heat shock to induce CreER expression in a restricted region of transgenic fish carrying a tissue-specific *loxP-DsRedx-loxP-GFP* reporter and a *hsp70l:mCherry-T2a-CreER*$^{T2}$. The removal of DsRedx cassette is permanently inherited so that the progenies derived from the targeted tissue will display GFP instead of DsRedx (*Xu et al., 2015*; *Tian et al., 2017*; *He et al., 2018*).

However, it has been realized that heterogeneity broadly exists in multiple cell populations during hematopoiesis (*Tian et al., 2017*; *Wilson et al., 2015*; *Chen et al., 2011*; *Crisan and Dzierzak, 2016*; *Ye et al., 2017*). The dissection of heterogeneity requires a lineage tracing strategy with single cell resolution. Yet, the previous IR-LEGO techniques (*Deguchi, 2009*; *Kamei et al., 2009*; *Shimada et al., 2013*; *Okuyama et al., 2013*; *Xu et al., 2015*; *Tian et al., 2017*; *Singhal and Shaham, 2017*; *He et al., 2018*; *Kawasumi-Kita, 2015*; *Suzuki et al., 2014*; *Eiji, 2013*; *Hayashi et al., 2014*; *Miao and Hayashi, 2015*; *Hasugata et al., 2018*) face the following fundamental challenges: (1) high-precision and efficient labeling of the targeted single cell; (2) fine balance between labeling efficiency and cell viability after heat shock treatment; (3) permanent marking and long-term tracing of all progeny of the labeled single cell; (4) rigorous statistical analysis to quantitatively determine the lineages of the labeled single cell under random basal interference. These challenges hamper the wider application of IR-LEGO for cell fate mapping, and it remains unclear whether this technique can indeed be used for long-term tracing of multiple lineages of a single multipotent cell, such as HSPC.

Besides IR-LEGO technique, other single-cell lineage tracing methodologies invented in past decade suffer from similar problems. Cell barcode techniques, either by retroviral library infection to insert inheritable DNA barcodes (*Naik et al., 2014*), or by CRISPR/Cas9 system to accumulate

random mutations (*McKenna et al., 2016*; *Kalhor et al., 2017*), have been used to perform single-cell lineage tracing in hematopoiesis studies (*Gerrits et al., 2010*; *Lu et al., 2011*). Single-cell RNA-sequencing is also a prevalent way to depict lineage hierarchy (*Hoppe et al., 2014*; *Zhou et al., 2016*; *Athanasiadis et al., 2017*). However, these non-imaging based techniques are not suitable for tracing the dynamic behaviors of the targeted cells and their progenies. A multicolor strategy, which stochastically expresses multiple fluorescent reporters in target cells via Cre-mediated recombination, forming dozens of different color modes to distinguish individual cells and their progenies, has been utilized for cell fate mapping (*Livet et al., 2007*; *Cai et al., 2013*). Despite its success in some zebrafish hematopoiesis study (*Henninger et al., 2017*; *Pan et al., 2013*), this multicolor labeling of cells makes it difficult to directly visualize the development of individual cell lineage. In addition, the lineage hierarchy could be misinterpreted when unrelated cells share the same color or Cre-mediated recombination occurs in the daughter cells. Likewise, photo-convertible protein or caged fluorescent dyes approaches also have limitation for long-term tracing due to the self-degradation and rapid dilution of fluorophores during cell division (*Tian et al., 2017*; *Warga et al., 2009*). An optical uncaging method was used to label a targeted cell and its progeny in zebrafish through Cre-based gene recombination (*Sinha et al., 2010a*; *Tekeli et al., 2016*). However, the basal uncaging level of the caged compound in zebrafish embryo was as high as 28% (*Sinha et al., 2010b*), limiting the application of this technique to trace the long-term development of highly dynamic cells, such as stem cells. Thus, developing a highly precise single-cell labeling method for the long-term in vivo tracing of individual cells will be important for understanding the heterogeneity of HSPCs.

To overcome the drawbacks of existing techniques, we develop a high-precision single-cell IR-LEGO technology, in which a two-photon fluorescent thermometer is utilized to measure the temperature rise in vivo to achieve precise single-cell labeling. Using this tool, we document that the hemogenic endothelium (HE) cells in the posterior blood island (PBI) of zebrafish are heterogeneous in terms of hematopoietic potential. Our study demonstrates that the high-precision single-cell IR-LEGO technology has outstanding capacity to perform single-cell labeling and long-term in-vivo lineage tracing.

## Results

### Single-cell IR-LEGO technology

A 1,342 nm diode-pumped solid-state (DPSS) IR laser is used as the heat-shock light source in our single-cell IR-LEGO heat-shock microscope system (*Figure 1A*). The laser at this wavelength provides an appropriate balance between the absorption efficiency of water and penetration depth in tissue (Appendix 1). The IR laser is integrated with a two-photon microscope, and it is guided by the two-photon fluorescence imaging to heat the targeted cell. A water-immersion objective with a large numerical aperture (NA) is used to generate highly localized and stable laser heating at different depths in tissues. The numerical simulation (*Figure 1—figure supplement 1* and Appendix 2) shows that heat shock of high spatial resolution can be achieved by generating a point heat source inside tissue, a medium of relatively poor thermal conductivity. A large temperature gradient can be created in the region of about 10 μm size around the heat source (*Figure 1—figure supplement 1D*), suggesting that the thermal energy produced by a highly focused IR laser heating in tissue could be confined within the single-cell dimensions for efficient single-cell gene induction.

Since the heat shock efficiency varies as the type and location of targeted cells, it is of great significance to develop a reliable method that can objectively determine the optimal IR laser heating conditions for single-cell gene induction. Although previous studies demonstrated that temperature-sensitive fluorescent proteins, such as GFP and mCherry, can be used as thermometers to estimate the temperature rise in cells induced by IR laser irradiation (*Kamei et al., 2009*; *Singhal and Shaham, 2017*), this single-molecular/one-color thermometry has been shown to produce significant errors, likely because of the fluctuation of excitation laser power, or the interference of complex microenvironment on signal intensities of fluorescence emission (*Estrada-Pérez et al., 2011*). In order to precisely characterize the heat diffusion from the highly focused IR laser, we developed a two-photon fluorescent thermometry (TPFT) technique to measure the three-dimensional (3D) distribution of temperature rise in the region close to the laser focal point in water, 3% agarose (a tissue phantom of thermal conductivity similar to typical tissues) (*Huang et al., 2004*) and live zebrafish,

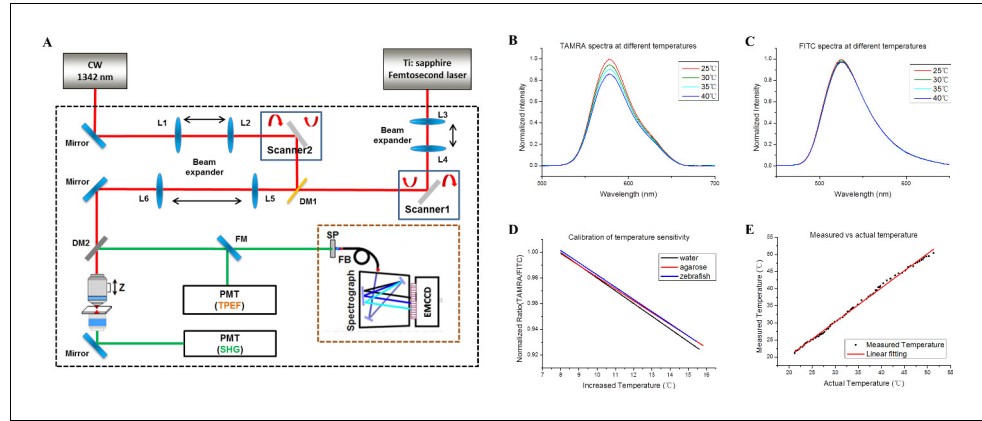

**Figure 1.** IR-LEGO heat-shock microscopy and fluorescent thermometry. (**A**) Schematic diagram of the integrated heat-shock microscope system and fluorescent thermometry. A 1,342 nm continuous-wave (CW) IR laser was used for localized heat shock, and a femtosecond laser was used for TPEF imaging and spectroscopy. The femtosecond laser and CW laser beams were individually controlled by two pairs of galvanometer scanners. DM: dichroic mirror; PMT: photomultiplier tube; FM: foldable mirror; SP: short-pass filter; FB: fiber bundle; EMCCD: electron multiplying charge-coupled device; L1-L6: relay lens; SHG: second harmonic generation signals. (**B, C**) TPEF spectra of TAMRA and FITC at different temperatures, respectively. (**D**) Calibration of temperature sensitivity: fluorescence intensity ratio (TAMRA/FITC) as a function of temperature in water, 3% agarose (tissue phantom) and zebrafish muscle in vivo. (**E**) Representative results of measured temperature vs. actual temperature in 3% agarose. Slope of linear fitting: 0.998.

The online version of this article includes the following source data and figure supplement(s) for figure 1:

**Figure supplement 1.** Simulation of heat diffusion from a point-heating source.

**Figure supplement 2.** Temperature sensitivity of fluorescent thermometry.

**Figure supplement 2—source data 1.** Statistics of temperature sensitivities of FITC and TAMRA fluorescence.

---

respectively. The thermometry measures the temperature rise in tissues noninvasively based on the fluorescence signals of two fluorescent dyes (Appendix 3) (*Estrada-Pérez et al., 2011*; *Natrajan and Christensen, 2008*). In details, a temperature-sensitive dye (*Figure 1*), tetramethylrhodamine (TAMRA) which is conjugated with dextran, is adopted as the probe dye in TPFT, while fluorescein (FITC), which is insensitive to temperature and also conjugated with dextran (*Figure 1C*), is used as a reference dye to eliminate the fluctuation of probe dye fluorescence caused by a variety of interferences (Appendix 3). The temperature dependencies of fluorescence measured in pure TAMRA and FITC solutions are -0.882 ± 0.100%/°C and -0.165 ± 0.098%/°C respectively (*Figure 1—figure supplement 2A and B*, *Figure 1—figure supplement 2—source data 1*). The two-photon excited fluorescence (TPEF) intensity ratio of TAMRA and FITC is linearly correlated with the solution temperature due to large difference in temperature sensitivity between two dyes (*Figure 1D* and *Figure 1—figure supplement 2*). The temperature sensitivities of fluorescence intensity ratio are similar in water solution, 3% agarose and zebrafish in vivo (*Figure 1D* and *Supplementary file 1a*), indicating that the temperature coefficient of the fluorescent dextran remains stable in different environments. The high consistency between the actual temperature and the measured temperature (*Figure 1E*) demonstrates that TPFT can be used as an effective tool for in vivo measurement of the local temperature rise induced by IR laser heating in tissues.

## Single-cell labeling in zebrafish

To study the dynamic change of temperature during IR laser heating, firstly we used TPFT to measure the temperature in water solution and 3% agarose with point heating. Low fluorescent dye concentrations were used to avoid self-absorption and fluorescence resonance energy transfer (FRET) (*Figure 2—figure supplement 1* and Appendix 4). Using a high-sensitivity EMCCD as the spectra detector, the dynamic temperature change at the heating site can be recorded in real time. We found that the temperature at IR laser focal point increased sharply within 1 ~ 2 s after point heating and remained stable over the exposure time of the IR laser, before decreasing quickly to the

ambient temperature as soon as the IR laser was turned off (*Figure 2—figure supplement 2A*). The depth of IR laser focal point in water and tissue phantom should be over 100 μm to minimize the thermal conduction at the intermedium surface (*Figure 2—figure supplement 2B and C*). Next, we measured the 3D temperature distributions (*Figure 2A and B*, *Figure 2—source data 1*), and found that high spatial resolution of heat shock could be achieved in 3% agarose. However, a large temperature gradient could not be built in water solution because of its high thermal conductivity and faster convection. Further, we conducted 3D temperature measurement in zebrafish in vivo and the results were compared with the measurement in agarose tissue phantom (*Figure 2C and D*, *Figure 2—source data 1*). Dextran-conjugated TAMRA and FITC were co-injected into fish embryos at one-cell stage. Then the embryos were raised to 1 day post fertilization (dpf). Muscle was chosen as the first tissue to perform temperature measurement because of its relatively uniform structure and simple microenvironment. To avoid laser-induced injury (*Figure 2—figure supplement 3*), we applied scan heating on zebrafish muscle and tissue phantom, in which the focused IR laser beam was scanned over an 8 μm × 8 μm region for 32 s instead of staying at a fixed heating point. Results showed that the thermal confinement in zebrafish muscle is higher than in tissue phantom, both laterally and axially (*Figure 2C and D*, *Figure 2—source data 1*). This indicates that the thermal conductivity of zebrafish muscle could be lower than that of 3% agarose. To visualize the thermal confinement clearly, we generated 3D view of the lateral temperature distributions based on the experimentally measured data (*Figure 2E*). As shown in *Figure 2 E1 and E2*, low thermal conductivity and inefficient convection of tissue phantom plays a critical role to confine the thermal energy and achieve single-cell resolution of heat shock. As shown in *Figure 2 E2 and E3*, there is no significant difference in the thermal confinement between point and scan heating methods. The results in *Figure 2 E3 and E4* demonstrate that TPFT can finely evaluate the thermal distribution in zebrafish muscle in vivo and paves the way for evaluation of single-cell heat shock in different zebrafish tissues.

Next, we examined the single-cell labeling efficiency in vivo through the single-cell IR-LEGO system in various kinds of cells, including myocytes in the skeletal muscle, neurons in the brain, and *coro1a*[+] leukocytes in the hematopoietic tissue at the aorta-gonad-mesonephros (AGM) and PBI of transgenic zebrafish (*Figure 2F–H*). Using TPFT, the temperature distribution in the heat shocked tissues was measured. With high IR laser power, the average temperature at the focal point (P00) of IR laser can reach as high as 50℃ (*Figure 2—figure supplement 4A*, *Figure 2—figure supplement 4—source data 1*). Although considerable success rates of overall cell labeling can be achieved with this high temperature heat shock, the percentages of single-cell labeling within the overall labeling are relatively low (*Supplementary file 1b*). This is due to that effective heat shock gene induction can be induced with environmental temperature higher than 38℃ (*Shoji and Sato-Maeda, 2008*) and the heating region of temperature over 38℃ was greater than the single cell size. Therefore, in order to increase the efficiency of single-cell labeling, the IR laser power was then optimized to restrain the heat diffusion and limit the effective area of gene induction in a single-cell dimension. With the optimized heat shock condition, the temperature at 10 μm away from the focal point (P10) dropped below 38℃ (*Figure 2—figure supplement 4B and C*, *Figure 2—figure supplement 4—source data 1*), preventing unwanted gene induction in neighboring cells. It was demonstrated that, after heat shock with optimized laser condition, successful overall cell labeling in myocytes, neurons and leukocytes can be observed in 36.7%, 18.6% and 50% of zebrafish, respectively, among which the efficiencies of single-cell labeling for these three types of tissues are 54.5%, 100% and 77.8%, respectively (*Figure 2—figure supplement 4D* and *Supplementary file 1b*). Compared with the high-temperature heat shock, the success rates of overall cell labeling with optimized heat shock condition were decreased, but the efficiency of single-cell labeling was significantly improved. This demonstrates that the single-cell IR-LEGO technology can efficiently and precisely induce heat shock-mediated gene editing within single cell in vivo in multiple tissues under optimized condition.

## Tracing single HE and progenies

Similar to mammals, the definitive hematopoiesis of zebrafish initiates from the HE in the ventral wall of the dorsal aorta and was thought to give rise to hematopoietic stem cells (HSCs) (*Bertrand et al., 2010*; *Kissa and Herbomel, 2010*; *Tian et al., 2017*). Yet, our recent study has shown that in addition to generating HSCs, the HEs in the aorta also produces non-HSC progenitors capable of differentiating into T cells, myeloid and erythroid lineages but not B cells in a transient manner

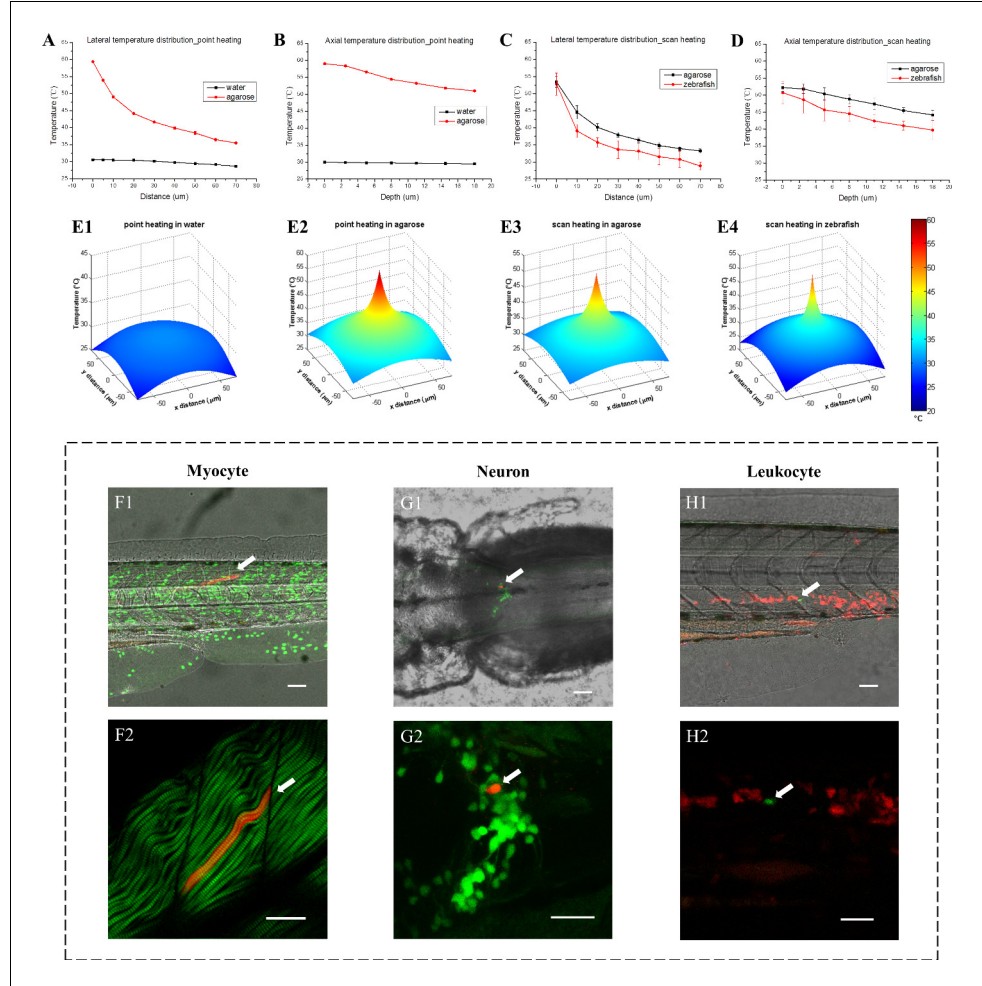

**Figure 2.** 3D temperature distribution measured through fluorescent thermometry and single-cell gene induction in zebrafish. (**A**) Lateral temperature distributions with IR laser heating in water and 3% agarose (95 mW IR laser was focused in samples for point heating). (**B**) Corresponding axial temperature distributions in water and 3% agarose. (**C**) Lateral temperature distributions with IR laser scan heating in 3% agarose and zebrafish muscle in vivo (95 mW IR laser was scanned in an 8 μm × 8 μm region during heating to avoid tissue injury). (**D**) Corresponding axial temperature distributions in 3% agarose and zebrafish muscle in vivo. Each statistical distribution curve in (**A–D**) is shown in terms of the mean with the standard deviation over more than five measurements. (**E1–E4**) 3D view of lateral temperature distributions with IR laser point and scan heating in water, 3% agarose and zebrafish muscle, respectively. (**F1**) A merged image of bright-field (gray) middle trunk, Dendra2-labeled nuclei (green) and a single myocyte expressing DsRedx via heat shock gene induction (red) in a one dpf zebrafish. (**F2**) A merged image of second harmonic generation (SHG) of muscle fibers (green) and the single myocyte expressing DsRedx (red). (**G1**) A merged image of bright-field (gray) hindbrain, GFP-labeled tyrosine hydroxylase-positive (th-positive) neurons (green) and a single neuron expressing DsRedx via heat shock gene induction (red) in a three dpf zebrafish. (**G2**) An enlarged image of (**G1**) with GFP and DsRedx-labeled neurons (by maximum projections). (**H1**) A merged image of bright-field (gray) middle trunk, DsRedx-labeled leukocytes (red) and a single cell expressing GFP via heat shock gene induction (green) in a two dpf zebrafish. (**H2**) An enlarged image of (**H1**) with merged DsRedx and GFP-labeled leukocytes. Arrows in (**F–H**): heat-shock labeled single myocyte, neuron and leukocyte, respectively. Scale bars: 50 μm (**F1–H1**); 30 μm (**F2–H2**).

The online version of this article includes the following source data and figure supplement(s) for figure 2:

**Source data 1.** 3D temperature distribution in water, tissue phantom and zebrafish during IR laser heat shock.
**Figure supplement 1.** Self-absorption and FRET of FITC and TAMRA.
**Figure supplement 2.** Temperature rise measured at focal point of IR laser under different heating conditions.
**Figure supplement 3.** Laser-induced injury in zebrafish muscle.
**Figure supplement 4.** Temperature distribution measurement and efficiency of single-cell labeling in different tissues of zebrafish.

*Figure 2 continued on next page*

*Figure 2 continued*

**Figure supplement 4—source data 1.** Temperature distributions under different heat shock conditions in three different tissues of zebrafish.

(*Tian et al., 2017*; *Bertrand et al., 2007*), highlighting the complexity of endothelial-hematopoietic transition (EHT), a process leading to the formation of blood stem and progenitor cells from the endothelium (*Bertrand et al., 2010*; *Kissa and Herbomel, 2010*). An important unsolved issue is whether these non-HSC-derived hematopoietic lineages, such as T lymphocytes, myeloid and erythroid cells, arise directly from distinct HE subpopulations that differentiate into different hematopoietic lineages independently or from a uniform HE population, which generates a common progenitor that further differentiates into multiple hematopoietic lineages.

To address the issue, we applied the high-precision single-cell IR-LEGO technology to single HE lineage tracing in the PBI region where all three non-HSC-derived hematopoietic lineages but not HSCs are generated (*Tian et al., 2017*; *Bertrand et al., 2007*). Specifically, we estimated the physical sizes of HEs by measuring the distance between the nuclei of neighboring endothelial cells, and it shows that the average length of endothelial cells along the aortic floor in the PBI is 24.9 μm, with the minimum of 11.2 μm (*Figure 3—figure supplement 1*, *Figure 3—figure supplement 1—source data 1*). Given that we have successfully constrained the effective heat-shock region (>38°C) within 10 μm along the ventral wall of aorta (*Figure 2—figure supplement 4B* and *Supplementary file 1b*), it is highly feasible to label HE at single-cell resolution using our heat-shock microscope system. In order to label HEs and follow their fates, we generated a double transgenic *Tg(kdrl:loxP-DsRedx-loxP-EGFP;coro1a:loxP-DsRedx-loxP-EGFP)* fish, in which editable genetic reporter loxP-DsRedx-loxP-EGFP is under the control of endothelial-specific *kdrl* promoter (*Jin et al., 2005*) and leukocyte-specific *coro1a* promoter (*Li et al., 2012*), thus HEs and leukocytes (including myeloid and lymphoid cells) were marked by DsRedx (*Figure 3A*). The double reporter transgenic line was then outcrossed with *Tg(hsp70l:mCherry-T2a-CreER^T2^)* fish to obtain a triple transgenic *Tg(kdrl:loxP-DsRedx-loxP-EGFP;coro1a:loxP-DsRedx-loxP-EGFP;hsp70l:mCherry-T2a-CreER^T2^)* line (referred to as 'triple Tg' hereinafter) (*Figure 3A*). In this triple Tg embryo, single-cell IR-LEGO system-induced heat-shock and 4-OH tamoxifen (4-OHT) treatment would induce and subsequently activate CreER within one HE, resulting in the excision of the DsRedx cassettes. As a consequence, the targeted HE and its leukocyte progenies would be distinguished from the unlabeled cells by EGFP expression (*Figure 3A*). To ensure only one single HE was labeled in each embryo, we irradiated the 4-OHT treated embryos at 26–28 hpf (*Figure 3B*) prior to the initiation of EHT (*Kissa and Herbomel, 2010*; *Tian et al., 2017*), then immediately imaged the heat-shocked embryos and control embryos continuously to 48 hpf (*Figure 3B and C*, *Videos 1* and *2*). As shown in *Figure 3C* and *Video 1*, GFP signal began to emerge at ~6 hrs post heat-shock and 4-OHT treatment. The embryos with single GFP⁺ HE during 20 hrs of time-lapse imaging were then selected for hematopoietic lineage analysis and the contribution of the labeled HE to T lymphocytes and myeloid cells in each of these embryos were determined by counting the coro1a:GFP⁺ cells at 7 dpf (*Supplementary file 1c and 1d*). Because T lymphocytes are strictly located in the thymus at 7 dpf, the GFP signals in thymus were analyzed to determine the numbers of T lymphocyte derived from the labeled HE. Whole-mount fluorescent in situ hybridization and antibody staining verified that the GFP⁺ cells in the thymus were indeed *rag1*⁺ T lymphocytes (*Figure 3—figure supplement 2A,B*). Additionally, co-labeling of thymic epithelium and different hematopoietic cell types using specific transgenic zebrafish lines exhibited that T

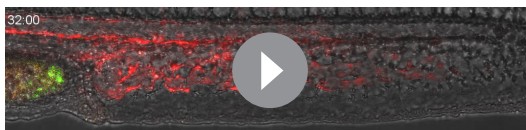

**Video 1.** The time-lapse live imaging of labeling a single caudal aorta endothelium cell in zebrafish.
https://elifesciences.org/articles/52024#video1

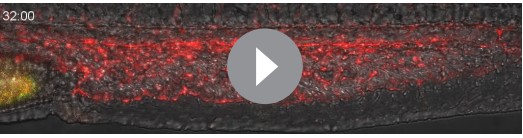

**Video 2.** The time-lapse live imaging of non-labeling control zebrafish.
https://elifesciences.org/articles/52024#video2

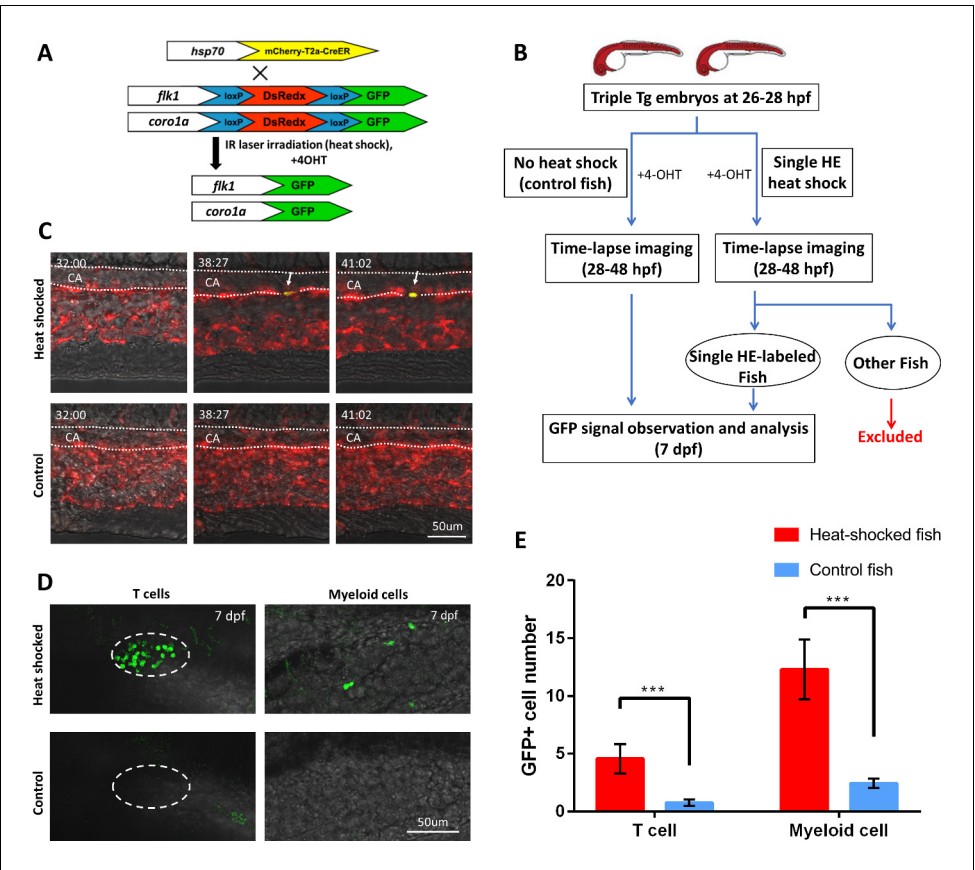

**Figure 3.** The high-precision IR-LEGO-mediated single HE lineage tracing. (A) The genetic labeling diagram of the lineage tracing experiment. The heat shock-induced CreER[T2] line *Tg(hsp70l:mCherry-T2a-CreER[T2])* is crossed with the double reporter fish *Tg(kdrl:loxP-DsRedx-loxP-EGFP;coro1a:loxP-DsRedx-loxP-EGFP)* to obtain triple transgenic *Tg(kdrl:loxP-DsRedx-loxP-EGFP; coro1a:loxP-DsRedx-loxP-EGFP; hsp70l:mCherry-T2a-CreER[T2])* embryo. Upon IR laser illumination, the targeted HEs will express CreER. After 4-OHT treatment, the CreER will enter the cell nucleus and remove the DsRedx cassettes flanked by loxP from the genome. After that, the target HEs and their hematopoietic progenies will express GFP signal, thus be distinguished from the unlabeled DsRedx[+] cells. (B) The work flow of the IR-LEGO-mediated single HE lineage tracing assay. (C) Representative time-lapse images of the single-HE heat-shocked embryos (upper row) and control embryos (lower row). Dotted lines depict the caudal aorta (CA) in the PBI region (dorsal wall on the top). Number at the top left corner in each image indicates the developmental stage of the embryos (hh:mm post fertilization). The heat-shock labeled HE (white arrow) on the ventral wall of CA gradually turns on the expression of GFP without affecting neighbor HEs, while the HEs in control embryos do not express GFP during time-lapse imaging. (D) Images of GFP[+] T cells and myeloid cells in single-HE labeled fish and control fish at 7 dpf (back on the top). The left column shows small and round coro1a: GFP[+] T cells in the thymus (depicted by dashed lines). The right column shows coro1a:GFP[+] myeloid cells on the trunk, which have irregular shape. While GFP[+] T cells and myeloid cells are persistently observed in many of the labeled fish (upper row), rare GFP signals are detected in most of the control fish (lower row). (E) Quantification of GFP[+] T cells and myeloid cells in both single HE-labeled fish (n=27) and control fish (n=109) at 7 dpf. Statistical analysis indicates that for both T cells and myeloid cells, the GFP[+] cell number in heat-shock labeled fish is significantly higher than that in control fish. The cell numbers are shown in terms of mean ± standard error of the mean. The Mann–Whitney–Wilcoxon rank-sum test was used for significance test. ***$P < 0.001$.

The online version of this article includes the following source data and figure supplement(s) for figure 3:

**Figure supplement 1.** Size of endothelial cells on the ventral wall of aorta in the PBI region.
**Figure supplement 1—source data 1.** Statistics of the size of endothelial cells on the aortic floor.
**Figure supplement 2.** Verification of T cells in the thymus of the single-HE labeled zebrafish at 7dpf.
**Figure supplement 3.** Verification of myeloid cells on the trunk of the single-HE labeled zebrafish at 7dpf.
**Figure supplement 3—source data 1.** The colocalization percentages of mpeg1[+], coro1a[+] and lyz[+] cells.

lymphocytes can also be effectively differentiated from other thymus-resident cells based on their small and round shapes (*Figure 3—figure supplement 2C–F*). Therefore, the numbers of GFP$^+$ T lymphocyte in thymus can be accurately counted as the contribution of the labeled HE cell to lymphoid lineage (*Figure 3D*). On the other hand, the GFP$^+$ cells distributed on the embryonic trunk are *lyz$^+$* or *mpeg1$^+$* myeloid cells (*Figure 3D*, *Figure 3—figure supplement 3*, *Figure 3—figure supplement 3—source data 1*). To avoid the disturbance by immature hematopoietic progenitors, GFP$^+$ cells in the hematopoietic tissues including AGM, caudal hematopoietic tissue (CHT) and kidney were excluded from the myeloid cell statistics. Results showed that, albeit a small portion of zebrafish show GFP$^+$ background signals in the control group (*Figure 3E*), the numbers of GFP$^+$ T lymphocytes and myeloid cells in the single HE-labeled group were significantly higher compared with that in the control group (*Figure 3E*), demonstrating that the high-precision single-cell IR-LEGO system can efficiently trace the progenies derived from a single HE with high fidelity.

Considering that the background signals may contribute to the GFP$^+$ cell numbers in the heat shocked zebrafish and interference their lineage interpretation, we applied the maximum likelihood estimation (MLE) method (*Shao, 2003*) to minimize the interference of background signal and to analyze the T lymphoid and myeloid potential of each single HE. The MLE method is a widely acknowledged statistical tool to extract desired information in the presence of noise background. It has been applied in the study of evolution, genetics and lineage tracing (*Tamura et al., 2011*; *Sousa and Hey, 2013*; *Excoffier and Heckel, 2006*; *Perié et al., 2014*; *Chan et al., 2019*; *Larsen et al., 2017*). For example, the MLE was used for robustly inferring evolutional trees in molecular evolutionary analysis (*Tamura et al., 2011*), and for population genomic inference with complex demographic models (*Sousa and Hey, 2013*; *Excoffier and Heckel, 2006*), and also to determine the cell lineage pathway by converting barcode relationships into a tree of cell division (*Perié et al., 2014*; *Chan et al., 2019*). In this study, the MLE method is used to calculate the lineage distributions that maximize the joint probability density of observed data in both single cell-labeled and control groups. Therefore, it can be used as an unbiased estimator to depict the lineages of a single HE. The details of the MLE model for HE lineage tracing were illustrated in Appendix 5. Results showed that 43.79% of the single HE labeled fish had both lymphoid and myeloid progenies, while 28.41% of the fish contained exclusively myeloid progenies (*Figure 4A,B* and *Supplementary file 1e, 1f*). Notably, 27.8% of the labeled fish showed neither GFP$^+$ T lymphocytes nor GFP$^+$ myeloid cells (*Figure 4A*), which could due to the fact that the labeled single HE in these fish was bona fide endothelial cells without hematopoietic potential, or contributed to other hematopoietic progenies such as erythroid lineage. Nevertheless, these data indicate that at least two distinct HE subpopulations exist in the aorta of the PBI: one population can give rise to both T lymphocytes and myeloid cells, while the other produces exclusively myeloid progenies (*Figure 4B, C*). This result demonstrates that combined with comprehensive statistical analysis, the high-precision single-cell IR-LEGO system is a powerful tool to perform in vivo single-cell fate mapping under unperturbed conditions.

## Discussion

Despite the contribution of the IR-LEGO heat shock technique in previous bulk fate mapping studies, its spatial resolution does not meet the requirement of single-cell lineage tracing (*Xu et al., 2015*; *Tian et al., 2017*; *He et al., 2018*; *Henninger et al., 2017*). In this work, we developed an advanced single-cell IR-LEGO microscope system equipped with a cutting-edge fluorescent thermometer. By associating temperature with the fluorescence intensity ratio, this dual-dye fluorescent thermometry is immune to the fluctuation of excitation laser power and may resistant to the micro-environmental variations. This advantage leads to a significantly higher signal-to-noise ratio of temperature measurement than the single-probe thermometry (*Deguchi, 2009*; *Kamei et al., 2009*). Unlike other temperature probes such as nanoparticles (*Alkahtani et al., 2017*; *Blakley et al., 2016*), the dextran-conjugated fluorescent dyes used in this TPFT are highly bio-compatible and are ideal options for local temperature probing in live animal models. Benefiting from the intrinsic 3D sectioning capability of two-photon optical microscope, high spatial-resolution (<1 μm laterally) temperature measurement can be achieved in tissues. With the equipment of a high-speed EMCCD to detect fluorescence spectra, the temporal resolution can be as high as 0.02 s, enabling real-time temperature monitoring during the IR laser heating. Another improvement of this system is the flexible

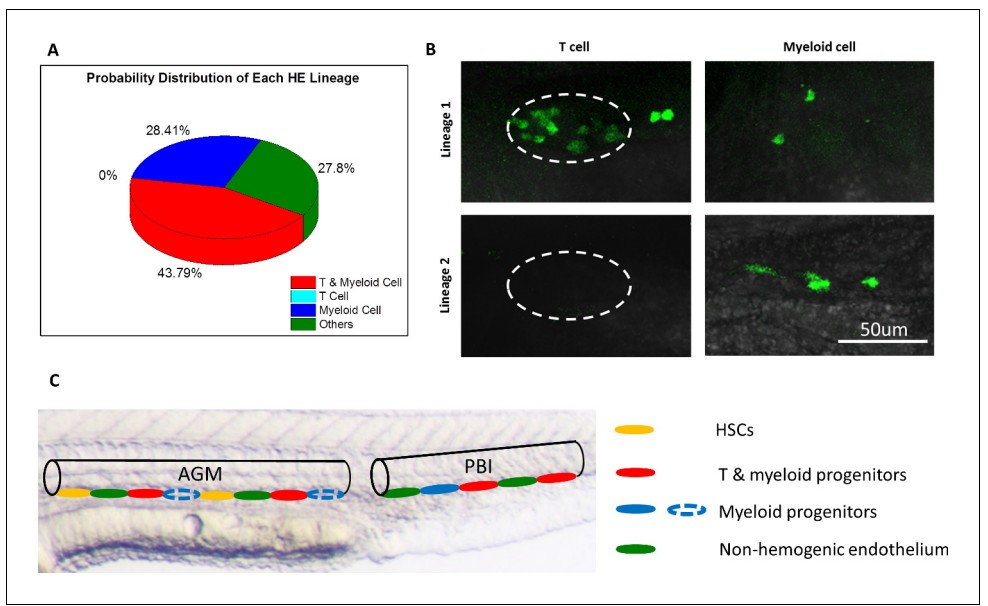

**Figure 4.** The heterogeneous hematopoietic lineages of HE. (**A**) Distribution of probability of each HE subpopulation. At least two subpopulations of HEs with distinct hematopoietic potentials exist in the PBI. One group of HEs has lymphoid and myeloid potential, while the other group generates myeloid lineage only. HEs with lymphoid lineage potential only were not found. The total number of single-HE labeled zebrafish is 27. (**B**) Representative images of GFP$^+$ T cells and myeloid cells derived from two distinct HE subpopulations. It shows that the myeloid-lymphoid bipotent HEs give rise to both T cells and myeloid cells (the upper row), while the myeloid unipotent HEs generate myeloid progenies exclusively (the lower row). The left column shows small and round coro1a:GFP$^+$ T cells in the thymus (depicted by dashed lines). The right column shows coro1a:GFP$^+$ myeloid cells on the trunk, which have irregular shape. (**C**) A schematic diagram illustrates the heterogeneity of hematopoietic potential of HEs. The aortic lumen in AGM and PBI are represented by black lines. The different HE subpopulations on the ventral floor of aorta are indicated by ellipses with different colors. The HEs in PBI give rise to T lymphoid-myeloid bipotent progenitors (red) and myeloid unipotent progenitors (blue), while the HEs in the AGM produce HSCs (orange) and T lymphoid-myeloid bipotent progenitors. It is possible that the AGM HEs can also generate myeloid unipotent progenitors (blue dotted ellipses), similar to their counterparts in the PBI.

The online version of this article includes the following source data and figure supplement(s) for figure 4:

**Figure supplement 1.** Assessment of HE cell damage caused by IR-LEGO heat shock.

**Figure supplement 1—source data 1.** Quantification of the HE cell death under different heat shock conditions.

---

control of heat shock modes. In the old IR-LEGO system used in our previous works (*Xu et al., 2015*; *Tian et al., 2017*; *He et al., 2018*), a loosely focused IR laser beam was fixed at a single spot of large region in tissues during heat shock. That system can achieve bulk cell labeling without induction of cell death (*Figure 4—figure supplement 1C*, *Figure 4—figure supplement 1—source data 1*), due to the large focal spot size (26 μm$^2$) and low power density of laser beam. In contrast, in current single-cell IR-LEGO microscope system, we adopted a high-NA objective to tightly focus the IR laser into an extremely small region (0.4 μm$^2$) inside a target cell, which is indispensable for high-resolution single-cell labeling. However, its high power density raises the possibility of cell damage for single-point heating mode (*Figure 2—figure supplement 3*, *Figure 4—figure supplement 1 A and B*, *Figure 4—figure supplement 1—source data 1*, *Video 3*). Therefore, to avoid overheating or photochemical damage, our current study utilizes two pairs of scanning galvo mirrors, which enable independent control of the IR laser beam and the fluorescence excitation laser beam, to perform two-dimensional scanning over the targeted cells. In this work, we

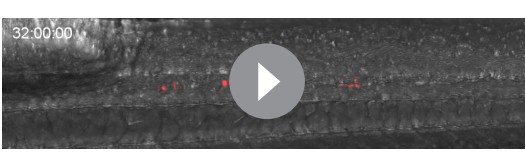

**Video 3.** The time-lapse live imaging of point heat shocked zebrafish.
https://elifesciences.org/articles/52024#video3

applied 32 s heating by constantly scanning the IR laser beam over an area of 8 μm × 8 μm in a cell. This scan-heating mode avoids the quick heat accumulation at single point and effectively reduces the cell damage (*Figure 4—figure supplement 1A and B*, *Figure 4—figure supplement 1—source data 1*, *Videos 4* and *5*). Additionally, this dual-scanner setup also expedites the characterization of temperature distribution during IR laser heating. Our results show that without strict thermal confinement, the efficiency of single-cell labeling would be very low and the heating effects in different tissues differ largely from each other (*Figure 2—figure supplement 4* and *Supplementary file 1b*). Therefore, optimization of heat shock condition to constrain thermal diffusion in specific tissues is of high necessity for high-throughput single-cell labeling and practical single-cell lineage tracing study.

Under optimized condition, our single-cell IR-LEGO technology can simultaneously achieve labeling, visualization and long-term tracing of single cell. Benefiting from this method of high spatial resolution and reliable fidelity, we uncovered the heterogeneity of HEs in the PBI of zebrafish and unveiled the complexity of lineage hierarchy in the definitive hematopoiesis. As a strategy for cell labeling and tracing, our single-cell IR-LEGO technique can be applied on any cell type as long as the IR laser can penetrate through the tissues above the targeted cells. In this work, we have demonstrated that this technique can precisely label single cell in different tissues with various depths and microenvironments, such as muscle, brain and hematopoietic tissue, indicating its great value for many other fields besides developmental biology.

It is noticed that the labeling efficiency of single-cell IR-LEGO depends on heat shock conditions and cell types. Our results show that the efficiency of labeling single myocyte and single leukocyte is 54.5% and 77.8%, respectively, while the success rate of single-neuron labeling can reach 100%, due to their lower density in the brain (*Figure 2—figure supplement 4D* and *Supplementary file 1b*). However, the efficiency of single HE labeling in the present lineage tracing assay is relatively low (29.3%; 27/92), mainly due to the strict criteria (live imaging for scoring single cell labeling) we set for scoring. Unlike other cell types, the HEs are highly mobile. Upon EHT, they quickly undergo cell division and differentiate into highly mobile hematopoietic precursors. As a consequence, a portion of heat-shocked single HE would not be counted due to their proliferation (more than one cells) and migration (lost in the circulation) before the appearance of GFP expression (induced by heat-shock). Thus, the actual efficiency of single HE labeling should be significantly higher than 29.3%. Even so, the 29.3% labeling efficiency is higher than that of previously reported single cell labeling in *Drosophila* (below 20% [*Miao and Hayashi, 2015*]). Although single cell labeling was reported in zebrafish by one study (*Eiji, 2013*), its efficiency was not discussed (the efficiency documented in that study refers to the overall efficiency of both single and multiple cells labeling). Moreover, none of these previous works has applied single cell labeling and long-term lineage tracing on highly mobile cells such as HEs. We believe that the single cell labeling strategy based on our newly developed IR-LEGO technology is the first comprehensive and integrated approach demonstrated for reliable and accurate single cell lineage tracing.

The background noise, namely the appearance of GFP signals in zebrafish without heat shock, creates challenge to single-cell lineage tracing. In fact, background noise is a common issue among various single-cell lineage tracing technologies, such as high-throughput techniques (*Yuan et al., 2017*) and optical tracing based on Cre-LoxP system (*Henninger et al., 2017*). For IR-LEGO technique, the background signals may arise from undesirable activation of the heat shock promoter which occurs occasionally and randomly during zebrafish development. This background noise is unavoidable, especially for long-term lineage tracing study. For bulk-labeling fate mapping, the background signals have little impact on the lineage interpretation because the number of labeled cells and their progenies are much larger than that of background signals. For single-cell lineage

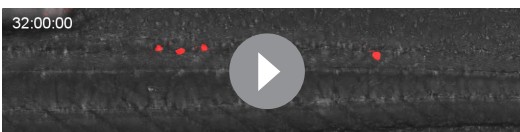

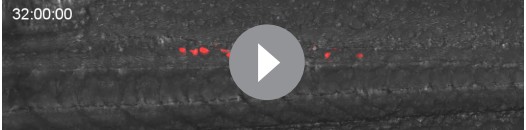

**Video 4.** The time-lapse live imaging of scanning heat shocked zebrafish.
https://elifesciences.org/articles/52024#video4

**Video 5.** The time-lapse live imaging of non-heat shocked control zebrafish.
https://elifesciences.org/articles/52024#video5

tracing, however, the number of progenies derived from a single target cell is limited, thus interference of background signals cannot be ignored. To minimize the effect of background noise, we applied the MLE method, a classic and well-established statistical tool, to analyze all the measurement data and estimate the lineage distribution. In fact, the MLE method has been widely applied in different aspects of biological studies including single cell transcriptomics, genome-wide association study and lineage analysis with DNA barcoding (*Chan et al., 2017*; *Austerlitz et al., 2009*; *Aulchenko et al., 2010*), showing that the successful integration of mathematic and biological methods can explicitly improve statistical power. For the MLE model in this study, each single HE-labeled zebrafish was classified into different lineage types based on the presence or absence of GFP$^+$ T/myeloid cells, regardless of the specific numbers of GFP$^+$ cells. In this way, the probabilities of each lineage calculated by the MLE method are not affected by the variations of GFP$^+$ cell numbers in both heat shocked and control zebrafish. Our results demonstrate that the MLE statistical method improves fidelity and broadens applicability of single-cell IR-LEGO system in the study of single-cell lineage tracing.

Our single-cell tracing study suggested that HEs in the PBI directly give rise to at least two distinct hematopoietic precursors: one capable of generating both T lymphocytes and myeloid cells and the other producing myeloid cells only. Although HEs in the PBI are also known to generate erythrocytes (*Tian et al., 2017*), our current study could not investigate T lymphoid, myeloid and erythroid lineage simultaneously because the coro1a promoter is leukocyte-specific. In principle, a possible solution is to generate *globin:loxP-DsRedx-loxP-GFP;coro1a:loxP-DsRedx-loxP-EGFP* double reporter line for triple lineage fate mapping analysis. However, a major drawback of this design is that incomplete gene editing, in which the heat shock-induced CreER edits one but not the other reporter cassette, may cause the misinterpretation of the linage potential. Given the fact that the erythroid-myeloid and lymphoid-myeloid progenitors but not lymphoid-erythroid progenitors have been identified in both mouse and zebrafish (*Bertrand et al., 2007*; *Perdiguero and Geissmann, 2016*; *McGrath et al., 2015*; *Böiers et al., 2013*), we attend to speculate that the HEs in the PBI may likely give rise to two different progenitor populations with lymphoid-myeloid potential and erythroid-myeloid potential respectively (*Figure 4C*). Studies in mice suggested that erythro-myeloid progenitors (EMPs) and HSCs are derived from distinct subpopulations of endothelial cells (*Chen et al., 2011*), and the mammalian HSCs showed heterogeneity during their emergence from the E11 AGM in mid-gestation embryos (*Ye et al., 2017*). In addition, our previous study has demonstrated that the generation of HSC-independent hematopoietic cells via EHT occurs in both PBI and AGM in zebrafish (*Tian et al., 2017*). Taken together, these studies raise the possibility that the heterogeneity of HEs is not a phenomenon restricted in the PBI region but broadly exists along the aortic floor (*Figure 4C*). However, we could not exclude the alternative possibility that the lineage biases among hematopoietic progenitors may be acquired by interacting with distinct niches. Indeed, studies in both mammals and zebrafish showed that new-born HSPCs dynamically interact with different cell types in various microenvironments, which is important for the migration, maintenance, proliferation and function of HSPCs (*Tamplin et al., 2015*; *Gao et al., 2018*; *Li et al., 2018*), and perhaps, is also crucial for lineage commitment. In the future study, it is of great interest to investigate whether the interactions between hematopoietic progenitors and niches contribute to the lineage heterogeneity.

# Materials and methods

**Key resources table**

| Reagent type (species) or resource | Designation | Source or reference | Identifiers | Additional information |
|---|---|---|---|---|
| Strain, strain background (*Danio rerio*) | *Tg(coro1a:loxP-DsRedx-loxP-EGFP)* | doi: 10.1016/j.devcel.2015.08.018. | | |
| Strain, strain background (*Danio rerio*) | *Tg(hsp70l:mCherry-T2a-CreER$^{T2}$)* | doi: 10.1182/blood-2014-09-601542 | | |

*Continued on next page*

*Continued*

| Reagent type (species) or resource | Designation | Source or reference | Identifiers | Additional information |
|---|---|---|---|---|
| Strain, strain background (*Danio rerio*) | *Tg(bactin2:loxP-STOP-loxP-DsRedx)* | doi: 10.1038/nature08738 | | |
| Strain, strain background (*Danio rerio*) | *Tg(kdrl:nls-EOS)* | doi: 10.1016/j.ydbio.2014.06.015 | | |
| Strain, strain background (*Danio rerio*) | *Tg(kdrl:loxP-DsRedx-loxP-EGFP)* | This paper | | Maintained in ZL. Wen lab |
| Antibody | Anti-GFP | Abcam | ab6658 | 1:400 Overnight 4°C |
| Antibody | donkey-anti-goat-488 secondary antibody | Invitrogen | A11055 | 1:400 Overnight 4°C |

## Heat-shock microscope and two-photon fluorescent thermometry

In the single-cell IR-LEGO heat shock microscope system (*Figure 1A*), a femtosecond Ti:sapphire laser (Chameleon Ultra II, Coherent, Santa Clara, CA) was used for the excitation of nonlinear optical (NLO) signals including TPEF and SHG. A DPSS low-noise CW IR laser (MLL-H-1342, Changchun New Industries Tech. Co,. Ltd.) at 1,342 nm wavelength was used for localized heat shock. The femtosecond laser beam was combined with the CW laser beam with a dichroic mirror (DMSP 1000, Thorlabs) and directed into a water-immersion objective (UAPON 40XW340, 1.15 NA, Olympus). Two pairs of galvanometer mirrors were used for x-y scanning of the femtosecond and CW laser beams, respectively. The objective was driven by an actuator for IR laser heating or NLO imaging at different depth. The backscattered NLO signals were collected by the objective and separated from the excitation light by a dichroic mirror (FF665-Di02, Semrock). In the fluorescent thermometry mode, the TPEF signals were focused into a spectrograph equipped with an EMCCD (DU-971N, Andor Technology), which enabled spectral analysis of the FITC and TAMRA fluorescence at a high resolution (0.4 nm). For the three-dimensional temperature profile measurement, a lens (L4, *Figure 1A*) on a linear translation stage (25 mm of travel) was moved along the light axis to change the focal plane of the femtosecond laser without changing the focus of the CW laser, allowing the measurement of temperature profile along the axial direction. In the imaging mode, the NLO signals were directed to a spectrograph via a round-to-line fiber bundle. The signals were analyzed by the spectrograph equipped with a linear array of 16 photomultiplier tubes (PMTs) and a time-correlated single photon counting (TCSPC) module (PML-16-C-0 and SPC-150, Becker and Hickl). Time-resolved NLO signals were recorded in 16 consecutive spectral bands with a 13 nm resolution, covering the spectral range from 450 nm to 645 nm simultaneously. Spectrally resolved images can be formed with a variety of NLO signals.

## Optical alignment of femtosecond laser beam and CW IR laser beam

Accurate co-localization of the focused femtosecond laser (830 nm) and IR laser (1,342 nm) beams is critical for the fluorescent thermometry and heat shock gene induction at single-cell resolution. Since IR laser wavelength is beyond the detection range of silicon-based detectors, an CCD camera can not be directly used for precise optical alignment of the probe beam (830 nm) and heat shock beam (1,342 nm). In this study, we painted a thin layer of black ink onto a coverglass and used it to identify the focal point of IR laser. In detail, when the cover glass was placed under objective and the IR laser power was appropriately controlled at low level, the black ink layer could only be vaporated by the laser at its focal point. The focal point without ink became a transparent spot of about 2 μm size that could be visualized in the bright field image captured by a CCD. This allows the focused beam positions of the 830 nm and 1,342 nm lasers to be visualized on the CCD camera simultaneously, to achieve a fine optical alignment of the two laser beams. The NLO imaging was then used to guide the IR laser to precisely aim at the targeted single cell for heat shock gene induction.

## Calibration of temperature sensitivity of FITC-TAMRA mixture

To calibrate the temperature sensitivity of FITC-TAMRA in water and tissue phantom (3% agarose), the individual or mixed 0.006% FITC (fluorescein and biotin-labeled dextran, 10,000 MW, Anionic, Lysine Fixable (Mini-Emerald), D-7178, Thermofisher Scientific) and 0.004% TAMRA (tetramethylr-hodamine and biotin-labeled dextran, 10,000 MW, Lysine Fixable (mini-Ruby), D3312, Thermofisher Scientific) solution was injected into a small homemade cuvette with two windows made of cover-glasses. The sealed cuvette was mounted in a petri dish filled with warm water through circulation via a water bath cabinet. The water temperature in petri dish was measured with a thermocouple, which was attached to the cuvette. The TPEF spectra of FITC-TAMRA mixture solutions were recorded at different temperatures controlled through the water bath cabinet. FITC and TAMRA fluorescence were decomposed using their individual spectra measured from pure dye solutions. After decomposition of the mixed spectra, the intensity ratios of FITC and TAMRA fluorescence were calculated to measure the temperature sensitivity.

To calibrate the temperature sensitivity of FITC-TAMRA mixture in zebrafish in vivo, the dextran-conjugated FITC and TAMRA were injected into zebrafish embryos at the single-cell stage (~1–2 nl/embryo). The embryos were raised to 1 dpf, 2 dpf and 3 dpf for the calibration of temperature sensitivity in muscle, AGM/PBI and hindbrain tissue, respectively. The zebrafish was mounted in 1% low-melting agarose and placed in an incubation system (Chamlide TC, Live Cell Instrument). A thermocouple was inserted into the agarose and close to the zebrafish to obtain the actual temperature. The FITC-TAMRA TPEF spectra were recorded at different environmental temperatures (25–38°C) controlled through the incubation system and used to measure the temperature sensitivities in corresponding tissues.

## Temperature distribution profiles with IR laser heating

We used the calibrated fluorescent thermometry to measure the local temperature rise induced by IR laser heating in water solution, tissue phantom (3% agarose) and zebrafish in vivo. Based on the calibrated temperature sensitivity, the local temperature rise was calculated by recording the changes in TPEF intensity ratios before and after IR laser heating. To measure the lateral tempera-ture distribution in water and tissue phantom, the 1,342-nm IR laser was fixed at the central position without scanning, while the 830-nm laser beam was scanned from the central position to the furthest distance of 70 μm away from the center to excite the TPEF of the FITC-TAMRA at different lateral positions. For the measurement of axial temperature distribution, the 830-nm probe beam was first co-localized with the 1,342-nm heat shock beam on the same focal plane and then separated axially from the heat shock beam by moving the lens (L4, *Figure 1A*) on a translation stage. For in vivo measurement of temperature distribution in the zebrafish, scan heating was performed to avoid laser-induced tissue injury. The 1,342-nm heating beam was scanned in an 8 μm × 8 μm region of the zebrafish tissues, while the 830-nm probe beam was scanned laterally and axially in the same way as in the water solution and tissue phantom to measure the temperature distributions. The 3D views of temperature distributions (*Figure 2E*) were plotted through the two-term Gaussian fitting of the discrete lateral temperature curves. The temperature is the sum of the environmental temper-ature (23°C) and the temperature rise measured through fluorescent thermometry.

## Zebrafish preparation and in vivo heat shock

For in vivo temperature measurement using fluorescent thermometry, dextran-conjugated FITC and TAMRA were injected into zebrafish embryos at the single-cell stage (~1–2 nl/embryo). The embryos were raised to the desired stages for fluorescent thermometry measurement.

For heat shock gene induction in muscle cells, CreER[T2] transgenic fish *Tg(hsp70l:mCherry-T2a-CreER[T2])* (*Hans et al., 2011*) were crossed with a *Tg(bactin2:loxP-STOP-loxP-DsRedx)* (*Bertrand et al., 2010*) reporter line. The embryos were injected with 1–2 nl PhOTO vector (*Dempsey et al., 2012*) at the single-cell stage, which labelled the cell membrane with cerulean and cell nuclei with Dendra2. The embryos were raised to 1 dpf and then mounted in 1% low-melting agarose for the heat shock experiment. With 1 μM 4-OHT treatment, scan heating was performed on myocyte nuclei for 32 s with the guidance of Dendra2 signals in TPEF imaging. The heat shock gene induction results were examined 24 hrs later by detecting the heat shock-induced DsRedx

expression in the myocytes. The SHG signal from the sarcomere of each muscle fiber was used to assist the validation of gene-induced cell numbers (*Figure 2 F2*).

For heat shock gene induction in tyrosine hydroxylase-positive (th-positive) neurons, *Tg(hsp70l:mCherry-T2a-CreER^{T2})* fish were crossed with reporter line *Tg(th:loxP-GFP-loxP-DsRedx)*. The embryos were raised to 3 dpf for the experiment. With 1 μM 4-OHT treatment, scan heat shock was performed on a single GFP-labeled th-positive neuron in an 8 μm × 8 μm area for 32 s in the hind-brain of zebrafish with the guidance of TPEF imaging. The heat shock gene induction results were examined 36 hr later by detecting the heat shock-induced DsRedx expression in the neurons.

For heat shock gene induction in leukocytes, *Tg(hsp70l:mCherry-T2a-CreER^{T2})* fish were crossed with reporter line *Tg(coro1a:loxP-DsRedx-loxP-EGFP)* (*Xu et al., 2015*). The embryos were raised to 2 dpf for the experiment. With 1 μM 4-OHT treatment, scan heat shock was performed on a single DsRedx-labeled leukocyte at the aorta-gonad-mesonephros (AGM) or the posterior blood island (PBI) region with the guidance of TPEF imaging. The scanning time was set as 32 s and scanning area at 8 μm × 8 μm to cover a single cell. After heat shock, live imaging was conducted to trace the migration of the heat-shocked cell and verify the heat-induced GFP expression in the target cells over the following 24 hrs.

For heat shock gene induction in hemogenic endothelium (HE) and the subsequent lineage tracing, we constructed transgenic fish *Tg(kdrl:loxP-DsRedx-loxP-EGFP)*, in which the blood vessel endothelium, including HEs, are labeled by DsRedx. To construct this transgene, the coro1a promoter in construct *coro1a:loxP-DsRedx-loxP-EGFP* was replaced by 6.5 kb vessel endothelial-specific *kdrl* promoter (*Cross et al., 2003*). The adult *Tg(kdrl:loxP-DsRedx-loxP-EGFP)* fish were crossed with *Tg(coro1a:loxP-DsRedx-loxP-EGFP)* fish to acquire double transgenic *Tg(kdrl:loxP-DsRedx-loxP-EGFP; coro1a:loxP-DsRedx-loxP-EGFP)* fish, in which both vessel endothelium and leukocytes are labeled. Then the adult double transgenic fish were crossed with *Tg(hsp70l:mCherry-T2a-CreER^{T2})* fish to acquire triple transgenic *Tg(kdrl:loxP-DsRedx-loxP-EGFP;coro1a:loxP-DsRedx-loxP-EGFP;hsp70l:mCherry-T2a-CreER^{T2})* fish (referred to as 'triple Tg' hereinafter). The triple Tg embryos were raised to 26–28 hpf for the experiment. With 1 μM 4-OHT treatment, scan heat shock was performed on a single *kdrl^+* HE on the ventral wall of caudal aorta at the PBI region with the guidance of TPEF imaging. The scanning time was set as 32 s and scanning area at 8 μm × 8 μm to cover a single HE. After heat shock, live imaging was conducted to trace the behavior of the heat-shocked HE and verify the heat-induced GFP expression in the target HEs till 48 hpf.

For the testing of cell damage/death caused by heat shock, *Tg(hsp70l:mCherry-T2a-CreER^{T2})* fish were crossed with reporter line *Tg(kdrl:nls-EOS)* (*Fukuhara et al., 2014*) to acquire double transgenic *Tg(hsp70l:mCherry-T2a-CreER^{T2};kdrl:nls-EOS)* fish. The double Tg embryos were raised to 26–28 hpf, and then the individual nls-EOS^+ HEs in the PBI region were exposed to UV laser to convert the EOS protein from green to red. Each converted fish was imaged immediately after photo-conversion to record the number and position of the converted cells. After that, a part of the converted embryos were treated with 4-OHT and the single red-EOS^+ HE was heat-shocked either by scanning over 8 μm × 8 μm area for 32 s or by single point irradiation for 32 s with the same laser power (80 mW). Rest of the converted embryos were only treated by 4-OHT as the control group. After heat shock, live imaging was conducted to record the cell death of the HEs in scanning heat shock group, single point heat shock group and control group, respectively. Cell death was validated based on two criteria. One is the disappearance of cells shortly after heat shock. The other is the observation that cell nuclei burst into fragments during live imaging. The same method was used to test the cell death caused by the old version of IR-LEGO system described previously (*Xu et al., 2015*). In that system, a doublet lens with 60 mm focal length was used as the objective lens to loosely focus the IR laser beam into samples. The converted embryos were treated with 4-OHT and single spot irradiation (80 mW) for 2 min was conducted on the PBI region. Then the cell death was assessed in the heat shocked group and the control group through live imaging.

## Live imaging of zebrafish after laser heat shock

Live imaging was performed according to the previous protocol with minor modifications (*Xu et al., 2016*). Embryos were mounted in 1% low-melting agarose and imaged on a Leica SP8 confocal microscope with a 25 °C thermal chamber. A 20x objective was used to take time-lapse images. The Z step size was set at 1.5 μm, with 20–30 planes in each z stack. For each embryo, images were taken every 20 minutes.

## Whole mount fluorescent in situ hybridization and antibody staining

The single HE-labeled fish were fixed on 7 dpf and then processed to whole mount in situ hybridization following a previously published protocol (*Thisse and Thisse, 2008*). Here, we made a few modifications of the protocol. First, we conducted an additional permeabilization step before proteinase K treatment by treating the samples with 100% acetone at −20℃ and then washing the samples in PBST for 5 min for 3 times, and we changed the anti-dig-AP antibody with anti-dig-POD and optimized the later fluorescent color reaction (TSA-cy3 system) steps according to another published protocol (*Welten et al., 2006*). For antibody staining, briefly, the samples were firstly blocked in 5% FBS in PBST, then incubated with goat-anti-GFP primary antibody (ab6658; Abcam) at 4℃ overnight. On the second day, the samples were washed in PBST for 30 min for 4 times, and then incubated in donkey-anti-goat-488 secondary antibody (A11055; Invitrogen) at 4℃ overnight. Finally, the samples were washed in PBST for 30 min. Of note, we inserted the antibody staining steps into the whole mount in situ hybridization steps by adding anti-dig-POD and goat-anti-GFP antibody at the same time, and then firstly completed antibody staining and finally went back to the rest of the whole mount in situ hybridization steps to finish color reaction for POD.

## Morphology characterization of thymus-resident cells

In order to confirm whether T lymphocytes can be distinguished from other cell types in thymus based on their morphology, we labeled and compared the size and morphology of potential thymus-residing cell types, including blood vessel endothelial cell, neutrophil, macrophage and T lymphocyte. Specifically, we crossed thymus epithelium-marking line *Tg(foxn1:mCherry)* with blood vessel endothelium-marking line *Tg(kdrl:EGFP)* or neutrophil-marking line *Tg(lyz:EGFP)* or macrophage-marking line *Tg(mpeg1:EGFP)* and imaged thymus regions of 7 dpf fish. In this way, we can observe the distribution of blood vessel endothelium, neutrophils and macrophages in thymus of 7 dpf fish (*Figure 3—figure supplement 2C–E*). To observe T lymphocytes in thymus, we crossed T lymphocyte-marking line *Tg(lck:loxP-DsRedx-loxP-EGFP)* with *Tg(hsp70I:mCherry-T2a-CreER$^{T2}$)* line and performed single spot IR laser illumination at 26 hpf PBI region as described previously to convert a small portion of T cells from DsRedx$^+$ into EGFP$^+$ (*Tian et al., 2017*). After IR laser illumination, the fish were treated with 4-OHT overnight, and were raised to 7 dpf for thymus imaging (*Figure 3—figure supplement 2F*). As shown in *Figure 3—figure supplement 2, T* cells can be effectively distinguished from other cell types in the thymus by their small and round shapes.

## Antibody staining, GFP-positive cell quantification and imaging

All the samples were directly fixed in 4% PFA at 4℃ overnight, then processed to whole-mount antibody staining as described elsewhere (*Barresi et al., 2000*). The primary antibody used in this study is anti-GFP antibody (ab6658, Abcam), and the secondary antibody is Alexa 488-anti-goat antibody (A11055, Invitrogen). After antibody staining, the zebrafish were mounted in 3% methylcellulose and the GFP-positive T lymphocytes as well as myeloid cells of each zebrafish were quantified manually under Nikon Eclipse Ti inverted fluorescent microscope. To capture the representative images of antibody stained samples, the zebrafish were mounted in 1% agarose and imaged with Leica SP8 confocal microscope.

## Statistical analysis

In the non-labeling zebrafish of the control group, the GFP$^+$ cell numbers are not in normal distributions. Therefore, a nonparametric test, the Mann–Whitney–Wilcoxon rank-sum test (also called the Mann-Whitney *U* test), was used for the significance test of the GFP$^+$ cells in the single HE-labeled and control zebrafish groups. The MLE method is used to calculate the lineage distributions of a single HE that maximize the joint probability density of observed data in both single cell-labeled and control groups. The details of the MLE model for HE lineage analysis were illustrated in Appendix 5.

## Acknowledgements

We thank Dr. Tao Yu for sharing the *Tg(th:loxP-GFP-loxP-DsRedx)* transgenic line. This work was supported by the National Key R and D Program of China through grant 2018YFA0800200, Hong Kong Research Grants Council through grants 662513, 16103215, 16148816, 16102518, T13-607/

12R, T13-706/11-1, AOE/M-09/12, T13-605/18W, C6002-17GF, and Hong Kong University of Science and Technology (HKUST) through grant RPC10EG33.

## Additional information

### Funding

| Funder | Grant reference number | Author |
|---|---|---|
| Hong Kong University of Science and Technology | RPC10EG33 | Jianan Y Qu |
| Research Grants Council, University Grants Committee | 662513 | Jianan Y Qu |
| Research Grants Council, University Grants Committee | 16103215 | Jianan Y Qu |
| Research Grants Council, University Grants Committee | 16148816 | Jianan Y Qu |
| Research Grants Council, University Grants Committee | 16102518 | Jianan Y Qu |
| Research Grants Council, University Grants Committee | T13-607/12R | Jianan Y Qu |
| Research Grants Council, University Grants Committee | T13-706/11-1 | Jianan Y Qu |
| Research Grants Council, University Grants Committee | AOE/M-09/12 | Jianan Y Qu |
| Research Grants Council, University Grants Committee | T13-605/18W | Jianan Y Qu |
| Research Grants Council, University Grants Committee | C6002-17GF | Jianan Y Qu |
| National key R&D Program of China | 2018YFA0800200 | Jin Xu |

The funders had no role in study design, data collection and interpretation, or the decision to submit the work for publication.

### Author contributions

Sicong He, Conceptualization, Data curation, Software, Formal analysis, Investigation, Methodology, Writing - original draft, Writing - review and editing; Ye Tian, Conceptualization, Data curation, Formal analysis, Investigation, Methodology, Writing - original draft, Writing - review and editing; Shachuan Feng, Conceptualization, Data curation, Formal analysis, Investigation, Methodology, Writing - original draft; Yi Wu, Data curation, Formal analysis; Xinwei Shen, Kani Chen, Formal analysis; Yingzhu He, Investigation; Qiqi Sun, Xuesong Li, Software, Methodology; Jin Xu, Conceptualization, Supervision, Funding acquisition, Methodology, Project administration; Zilong Wen, Jianan Y Qu, Conceptualization, Supervision, Funding acquisition, Writing - original draft, Project administration, Writing - review and editing

### Author ORCIDs

Sicong He (iD) https://orcid.org/0000-0002-0399-3904
Ye Tian (iD) http://orcid.org/0000-0002-9655-7123
Shachuan Feng (iD) http://orcid.org/0000-0001-8789-194X
Yingzhu He (iD) http://orcid.org/0000-0002-2416-6254
Jin Xu (iD) http://orcid.org/0000-0002-6840-1359
Jianan Y Qu (iD) https://orcid.org/0000-0002-6809-0087

### Decision letter and Author response

Decision letter https://doi.org/10.7554/eLife.52024.sa1

Author response https://doi.org/10.7554/eLife.52024.sa2

## Additional files

**Supplementary files**

• Source code 1. Source code of maximum likelihood estimation.

• Supplementary file 1. Tables. (a) Temperature sensitivity of fluorescent thermometry. (b) Temperature distribution with IR laser heat shock in zebrafish in vivo and efficiency of cell labeling in zebrafish tissues. (c) Numbers of GFP⁺ T lymphocytes and myeloid cells in each single HE-labeled zebrafish at seven dpf stage. (d) Numbers of GFP⁺ T lymphocytes and myeloid cells in each non-labeling control zebrafish at seven dpf stage. (e) The estimates and the corresponding 95% asymptotic confidence intervals of the probability for each hemogenic endothelium (HE) lineage in the single-HE labeled group. (f) The estimates and the corresponding 95% asymptotic confidence intervals of the probability for each type of zebrafish in the control group.

• Transparent reporting form

### Data availability

All data generated or analysed during this study are included in the manuscript and supporting files. Source data files have been provided for Figures 2-4, Figure 1-figure supplement 2, Figure 2-figure supplement 4, Figure 3-figure supplement 1, Figure 3-figure supplement 3, and Figure 4-figure supplement 1.

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

## Appendix 1

### Wavelength of IR Laser for Heat Shock

An IR laser was used as a heat shock light source because the absorption coefficient of water at 1,300–1,500 nm is about 1000 times higher than in the visible spectrum. According to the Beer-Lambert law (1-1), the attenuation of light is related to its optical depth and absorption in media. For in vivo heat shock of zebrafish, the absorption media include the immersion medium of the objective, cover glass, agarose and zebrafish tissues (**Appendix 1—figure 1**). We therefore apply a modified Beer-Lambert law (1-2) in multiple media where the total transmittance is associated with the absorbance of each medium.

$$T = \frac{P}{P_0} = e^{-\mu b} = 10^{-\varepsilon bc} = 10^{-(\mu/ln10)b} \ (\text{Beer' law}) \tag{1-1}$$

$$T = \prod e^{-\mu b} = e^{-\sum \mu b} \ (\text{modified Beer's law}) \tag{1-2}$$

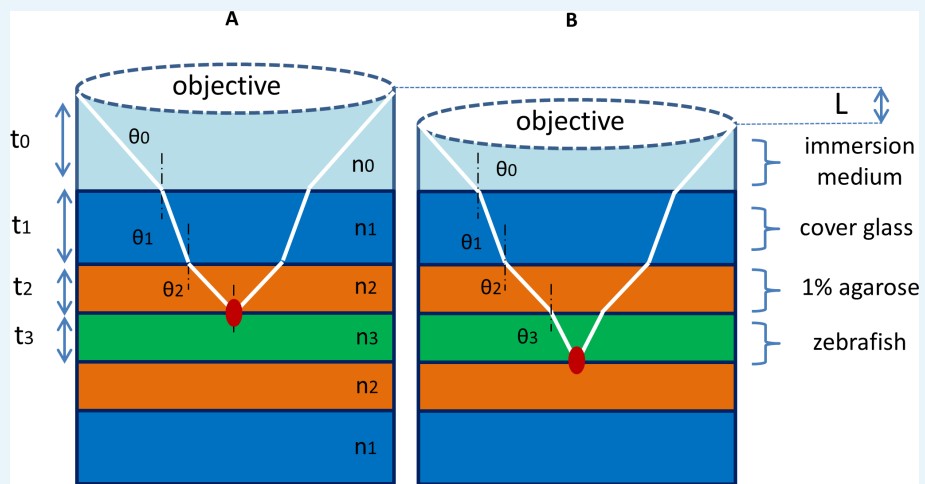

**Appendix 1—figure 1.** Multiple-media model for IR laser heat shock. (**A**) For IR laser heat shock in vivo, there are multiple media along the optical path from the objective to the focal point of IR laser, including the immersion medium (water), cover glass, agarose and zebrafish tissues. The total absorbance is determined by the absorption coefficient and thickness of each medium. (**B**) When the IR laser is focused at different depths in the zebrafish tissues, the absorbance along the optical path changes with the alteration of thickness of the immersion medium and tissues. A water immersion objective can stabilize the heating effect at different depths in tissue because of the reduced mismatch of absorption coefficients between immersion medium and tissue.

We compared two previously used IR wavelengths, 1,342 nm and 1,480 nm, for heat shock experiments (**Kamei et al., 2009**; **Xu et al., 2015**). The absorption coefficients of water at 1,342 nm and 1,480 nm are 3.1/cm and 25.7/cm respectively (**Palmer and Williams, 1974**). Although the absorption coefficient at 1,480 nm is much higher than that at 1,342 nm, the disadvantage with using a longer wavelength is that a considerable percentage of power is absorbed along the penetration path in media. A large amount of heat is produced out of focal region, which reduces the effective penetration depth and sacrifices the thermal resolution of heat shock. Quantitatively, we compared the laser power absorbed along the pathway in media at the wavelengths of 1,342 nm and 1,480 nm.

Assume the absorption length in the focal volume is about 10 μm, close to the size of a biological cell, Beer's law shows

$$absorption{:}A = 1 - T = 1 - e^{-\mu b}$$

$$\text{absorbed power at focal point:} P = P_0 * T_{path} * A_{focus} ,$$

$$\text{where } T_{path} = e^{-\mu b_p}, \ A_{focus} = 1 - e^{-\mu b_f} .$$

Based on the approximation that the length of the optical path in media is close to the working distance of the objective, we have

$$b_p = 250 \ \mu m \text{ and } b_f = 10 \ \mu m .$$

To achieve similar heating effect, the absorbed energy at the focal point with 1,342 nm heating should be close to that with 1,480 nm heating. Therefore,

$$P_{1342} * e^{-3.1*250*10^{-4}} * \left(1 - e^{-3.1*10*10^{-4}}\right) = P_{1480} * e^{-25.7*250*10^{-4}} * \left(1 - e^{-25.7*10*10^{-4}}\right)$$

$$P_{1342} = 4.67 * P_{1480} .$$

Compare the absorption along the light pathway:

$$\text{For } 1,342 \ nm\text{: } P_{1342} * \left(1 - T_{path}\right) = 0.075 P_{1342} = 0.35 P_{1480}$$

$$\text{For } 1,480 \ nm\text{: } P_{1480} * \left(1 - T_{path}\right) = 0.474 P_{1480}$$

$$\text{Difference of absorption in optical path: } \frac{0.474 - 0.35}{0.35} = 35.4\% .$$

It can be seen that 47.4% of the 1,480 nm laser power is absorbed by media before arriving at the focal point, compared with 7.5% for 1,342 nm laser. Furthermore, 35.4% more energy is absorbed by the defocused volume in the case of 1,480 nm heating, which reduces the spatial resolution of the IR laser heat shock. A 1.342 nm IR laser therefore provides a more appropriate balance between the absorption coefficient and penetration depth as well as energy utilization for the heat shock system.

**Appendix 2**

## Simulation of Heat Diffusion from a Point Heating Source

We used a finite-difference method to simulate the heat diffusion in tissues (**Holman, 2010**). In the finite-difference model, IR laser heating generates a point source in the tissue and spreads heat in the 3D space. To simplify the model, the highly focused IR laser beam is equalized as a single infinite heat source at the focal point of a large NA objective. The heat diffusion space is divided into equal increments in both the x and z directions (**Figure 1—figure supplement 1A**). It is assumed that the temperature of each nodal point in this space is determined by the temperatures of the four surrounding nodal points (**Figure 1—figure supplement 1C**). In the steady-state condition, the heat flow into any node is zero. For the finite-difference approximation of the nodal point's model, the heat equation can be expressed as

$$\frac{T_{m+1,n} + T_{m-1} - 2T_{m,n}}{(dx)^2} + \frac{T_{m,n-1} + T_{m,n-1} - 2T_{m,n}}{(dz)^2} = 0.$$

At the focal point, we take heat source into account:

$$\frac{T_{m+1,n} + T_{m-1} - 2T_{m,n}}{(dx)^2} + \frac{T_{m,n-1} + T_{m,n-1} - 2T_{m,n}}{(dz)^2} + \frac{q}{k} = 0.$$

where $q$ is thermal intensity and $k$ is thermal conductivity.

In the numerical analysis, the heat equation is written for each nodal point. The solved equations reveal the temperature distribution in the space defined by $L_x$ and $L_z$ (**Figure 1—figure supplement 1B**). As shown in **Figure 1—figure supplement 1A**, $L_x$ and $L_z$ are the dimensions of space along the x and z directions. Since the working distance of the objective is 0.25 mm, we assume the distance between the focal point and the interface of medium to be 0.1 mm ($L_z$ = 100 μm). The temperature at the interface of medium is assumed to be 25°C. The thermal conductivity of the tissue model is set as 0.5 W/mK, a typical thermal conductivity value of biological tissue (**Valvano et al., 1984**). The simulated temperature distributions with different $L_x$ values (**Figure 1—figure supplement 1D**) show that the approximation error of the boundary temperature is significant when $L_x$ is less than 150 μm. However, when $L_x$ is greater than 200 μm, the temperature distribution curves show similar profiles, with boundary temperature close to room temperature. The two-dimensional temperature distribution in the simulation space (**Figure 1—figure supplement 1B**) demonstrates that the thermal energy is well confined in a small volume, indicating that a high spatial resolution can be achieved by IR laser heating.

**Appendix 3**

## Two-dye Fluorescent Thermometry

Fluorescent thermometry is a noninvasive temperature measurement tool based on two fluorescent dyes with largely different temperature sensitivities. The temperature-sensitive property of fluorescent molecule results from the variation in its fluorescence quantum yield induced by the change of surrounding temperature (**Ko et al., 2006**). In principle, the fluorescence intensity is determined as follows:

$$\mathrm{I} = I_0 * C * \phi * \varepsilon \,,$$

where I is the fluorescence energy (W/m̃3); $I_0$ is the incident light flux (W/m̃2); C is the concentration of dye (kg/m̃3); $\phi$ is the quantum efficiency; and $\varepsilon$ is the absorption coefficient (m̃2/kg). Of these parameters, the quantum efficiency is usually temperature-sensitive. Two-dye fluorescent thermometry eliminates the variations of incident excitation light flux by measuring the fluorescence intensity ratios of two dyes with different temperature sensitivities. The ratio of the fluorescence intensities from two dyes is presented as follows:

$$\frac{I_A}{I_B} = \frac{C_A * \phi_A * \varepsilon_A}{C_B * \phi_B * \varepsilon_B} = K * \frac{\phi_A}{\phi_B} \,.$$

Let the temperature sensitivities of the two dyes be α and β (α, β >0), respectively. The quantum efficiencies of two dyes can be written as:

$$\phi_A = \phi_{A0} * (1 - \alpha\Delta T)$$
$$\phi_B = \phi_{B0} * (1 - \beta\Delta T) \,,$$

where $\phi_{A0}$ and $\phi_{B0}$ are the quantum efficiencies of the two dyes at room temperature and $T$ is the difference between the local temperature and room temperature (**Estrada-Pérez et al., 2011**; **Kuzkova et al., 2014**).

Then the temperature dependence of the fluorescence intensity ratio can be derived:

$$\frac{d\left(\frac{I_A}{I_B}\right)}{dT} = K * \frac{d\left(\frac{\phi_A}{\phi_B}\right)}{dT} = K\frac{\phi_A'\phi_B - \phi_A\phi_B'}{\phi_B^2} = K * \frac{\phi_{A0}(\beta - \alpha)}{\phi_{B0}(1-\beta T)^2} = \frac{I_{A0}}{I_{B0}}\frac{\beta - \alpha}{(1-\beta T)^2}$$

where $I_{A0}$ and $I_{B0}$ are the fluorescence intensities of the two dyes at room temperature. Since the temperature sensitivity of a dye is small and usually less than 1%/°C, then

$$\gamma = \frac{\beta - \alpha}{(1 - \beta\Delta T)^2} \approx \beta - \alpha$$

$$\frac{I_A}{I_B} = \frac{I_{A0}}{I_{B0}}(1 + \gamma\Delta T) \,.$$

Therefore, the fluorescence intensity ratio of the two dyes has an approximately linear relationship with the local temperature. In the case of β≪α or β≫α, which means the temperature sensitivities of the two dyes differ significantly, the fluorescence intensity ratio becomes highly dependent on temperature change. The theoretical temperature sensitivity of (1+γΔT) can be calculated based on the measurement of the temperature sensitivities of the two dyes used in this study (-0.165%/°C for FITC and -0.882%/°C for TAMRA) (**Figure 1—figure supplement 2A, B**). The simulation results show that the temperature sensitivity curves exhibit greater linearity in the case of β≪α than that of β≫α (**Figure 1—figure supplement 2C, D**). This means that the intensity ratio of TAMRA/FITC is more linearly dependent on temperature than the intensity ratio of FITC/TAMRA. It is therefore practical to measure the temperature from the fluorescence intensity ratio of TAMRA/FITC.

**Appendix 4**

## Self-absorption and FRET of FITC and TAMRA

The self-absorption and fluorescence resonance energy transfer (FRET) are two major factors to affect the measurement accuracy of fluorescent thermometry. In this study, we evaluated these two factors in FITC and TAMRA solutions of different concentrations.

## Self-absorption in FITC and TAMRA solutions

Firstly, the spectra measured from individual 1% FITC and TAMRA solutions show that the TPEF intensity of FITC and TAMRA decreased with increased depth (*Figure 2—figure supplement 1 A3, B3*). Meanwhile, the peak wavelength of the TPEF spectra demonstrated red shift with increased depth (*Figure 2—figure supplement 1 A1, B1*), which indicates that self-absorption occurred in the FITC and TAMRA solutions of concentration over 1%.

## FRET in mixed FITC and TAMRA solution

To verify the FRET, 1% FITC and TAMRA were mixed and the TPEF spectrum of the mixed solution was recorded at different depths (*Figure 2—figure supplement 1 C1*). The individual FITC and TAMRA spectra were then reconstructed by decomposing the spectra of FITC-TAMRA mixture using the method of least squares. The decomposed TPEF intensity of the FITC showed that the intensity of FITC decreased with increased depth and the slope of the decay curve was much greater than that of the individual 1% FITC solution (*Figure 2—figure supplement 1 D1*). This means that the FITC fluorescence was influenced not only by self-absorption but also by the FRET between the FITC and TAMRA molecules. On the other hand, the decomposed TPEF intensity of the TAMRA remained stable as the depth increased (*Figure 2—figure supplement 1 D1*). This can be explained by the FRET, in which the self-absorption of the TAMRA fluorescence was compensated for by the energy transfer from the FITC fluorescence.

## Minimization of self-absorption and FRET using low dye concentrations

Since the self-absorption and FRET are proportional to the dye concentrations, the FITC and TAMRA concentrations must be kept low. We found that the self-absorption of FITC and TAMRA fluorescence became negligible when the concentration of dye mixture was as low as 0.005% (*Figure 2—figure supplement 1 A2, B2, A3, B3*). Additionally, the FRET of the two molecules is also eliminated at these low-concentration levels (*Figure 2—figure supplement 1 C2*). The results show that the decomposed TPEF intensities of FITC and TAMRA remain stable with increased depth (*Figure 2—figure supplement 1 D2*). Meanwhile, the fluorescence intensity ratio is also independent of the depth (*Figure 2—figure supplement 1 C3*), so the variation of intensity ratio is only associated with the local temperature rise induced by IR laser heating. Thus, self-absorption and FRET have little influence on the calibration of temperature sensitivity in zebrafish in vivo as long as the concentration of FITC-TAMRA mixture in zebrafish is at 0.005% level.

**Appendix 5**

## Maximum Likelihood Estimation for HE Lineage Analysis

Here we elaborate the statistical method used to analyze the distribution of the possible lineages of HSC-independent hematopoietic progenitors.

The maximum likelihood estimation (MLE) method is an inference technique in which the estimated parameters are those that maximize the probability of the observed data (*Shao, 2003*). It has been widely used to unbiasedly estimate the specific parametric probability under the interference from irrelevant information. In this study, regarding to the lymphoid/myeloid lineages, a single progenitor cell has four potential fates: it may differentiate into (1) both T lymphocytes and myeloid cells; (2) T lymphocytes only; (3) myeloid cells only; (4) other cells rather than T lympochytes and myeloid cells. To analyze the distribution of each lineage, the embryos of the triple Tg line were heat shocked to label a single HE cell at the PBI (Group A in *Appendix 5—figure 1*). The single-HE labeled embryos were raised to seven dpf and GFP signals were examined. According to the appearance of GFP$^+$ T lymphocytes and GFP$^+$ myeloid cells, the seven dpf zebrafish were classified into four sub-groups: Group A-1 contained both GFP$^+$ T lymphocytes and GFP$^+$ myeloid cells; Group A-2 contained GFP$^+$ T lymphocytes only; Group A-3 contained GFP$^+$ myeloid cells only; Group A-4 contained neither GFP$^+$ T lymphocytes nor GFP$^+$ myeloid cells. Similarly, the control group (Group B) which was not heat shocked can also be separated into four subgroups from Group B-1 to Group B-4, as shown in *Appendix 5—figure 1*. The GFP$^+$ cells in Group B may be induced by the leaky of CreER expression during the development of progenitors/lymphocytes/myeloid cells. This basal GFP$^+$ expression may contribute to the total GFP$^+$ cell number in the heat shocked zebrafish and affect the distribution of each subgroup in Group A. Based on the statistical data in Group A and B (*Supplementary file 1c* and *Supplementary file 1d*), the MLE method was applied to analyze the genuine distribution of each lineage of progenitor cells that maximizes the joint probability density of data in both Group A and B.

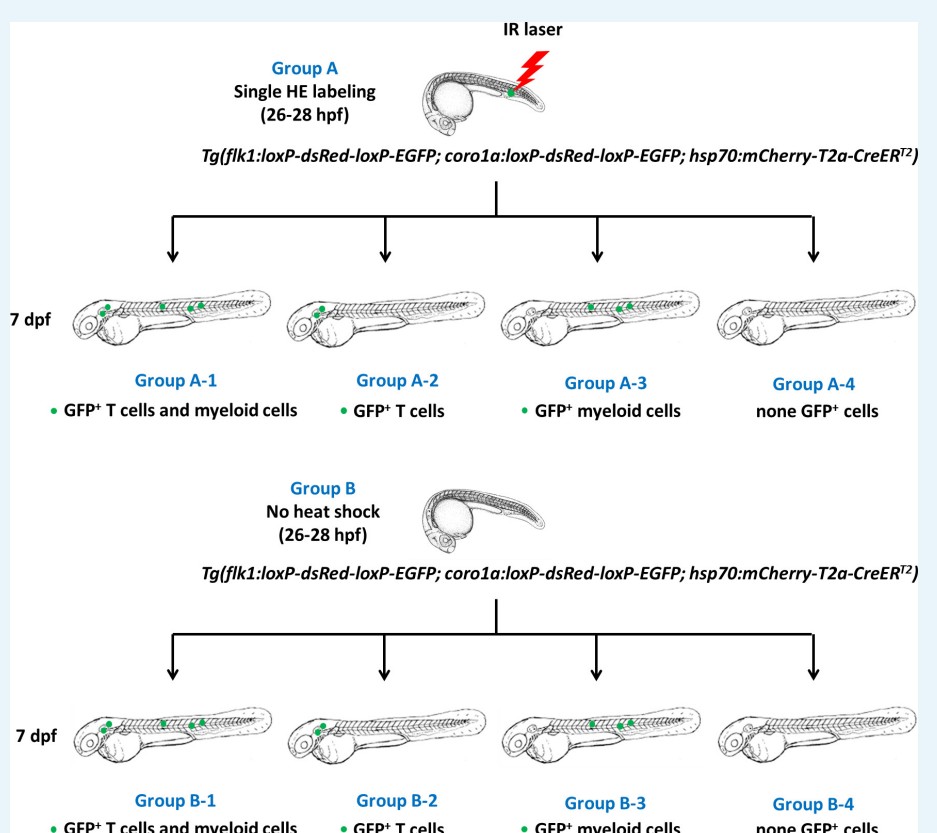

**Appendix 5—figure 1.** Classification for different subgroups of zebrafish in the single HE-labeled group and control group. In Group A, a single HE at the PBI of triple Tg embryo was labeled through IR-LEGO heat shock, while no heat shock was performed in Group B. The embryos were raised to seven dpf and divided into four subgroups according to the distribution of GFP$^+$ cells. Group A-1 and Group B-1 contained both GFP$^+$ T lymphocytes and GFP$^+$ myeloid cells. Group A-2 and Group B-2 contained GFP$^+$ T lymphocytes only. Group A-3 and Group B-3 contained GFP$^+$ myeloid cells only. Group A-4 and Group B-4 contained no GFP$^+$ cells. Each subgroup of zebrafish corresponds to a specific lineage of HSC-independent hematopoietic progenitors.

## Notations

Let $X = (X_T, X_M)^\top$ be the sign of the true number of T or M (myeloid) cells in a zebrafish in Group A, $\varepsilon = (\varepsilon_T, \varepsilon_M)^\top$ be the sign of the number of T or M cells in a zebrafish in Group B which we consider as noise, and $Y = X + \varepsilon = (Y_T, Y_M)^\top$ be the sign of the observed number of T or M cells in a zebrafish in Group A. Then the sample space of $X$, $\varepsilon$ and $Y$ is { (+, +), (+, 0), (0, +),(0,0)}.

## Estimated distribution of X

We assume the probability distribution of $X$ and $\varepsilon$ as in **Appendix 5—table 1**.

**Appendix 5—table 1.** The probability relationship among $X$, $\varepsilon$ and $Y$.

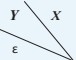

| | | (+,+) | (+,0) | (0,+) | (0,0) |
|---|---|---|---|---|---|
| | Prob. | $p_{11}$ | $p_{10}$ | $p_{01}$ | $p_{00}$ |
| (+,+) | $q_{11}$ | (+,+) | (+,+) | (+,+) | (+,+) |
| (+,0) | $q_{10}$ | (+,+) | (+,0) | (+,+) | (+,0) |
| (0,+) | $q_{01}$ | (+,+) | (+,+) | (0,+) | (0,+) |
| (0,0) | $q_{00}$ | (+,+) | (+,0) | (0,+) | (0,0) |

where $p_{11} + p_{10} + p_{01} + p_{00} = 1$ and $q_{11} + q_{10} + q_{01} + q_{00} = 1$. Then the distribution of $Y$ is shown in **Appendix 5—table 2**.

**Appendix 5—table 2.** The probability distribution of $Y$.

| Type | 1 | 2 | 3 | 4 |
|---|---|---|---|---|
| Y | (+,+) | (+,0) | (0,+) | (0,0) |
| Prob. | $p_{11}q_{00}+p_{11}q_{01}+p_{10}q_{01}$ $+p_{11}q_{10}+p_{01}q_{10}+p_{11}q_{11}+p_{10}q_{11}+p_{01}q_{11}+p_{00}q_{11}$ | $p_{10}q_{00}+p_{00}q_{10}+p_{10}q_{10}$ | $p_{01}q_{00}+p_{00}q_{01}+p_{01}q_{01}$ | $p_{00}q_{00}$ |

Thus, the likelihood function of the full data is

$$
\begin{aligned}
L = {} & q_{00}^{\#\varepsilon_{00}} q_{01}^{\#\varepsilon_{01}} q_{10}^{\#\varepsilon_{10}} q_{11}^{\#\varepsilon_{11}} \cdot \\
& (p_{00}q_{00})^{\#y_{00}} (p_{01}q_{00} + p_{00}q_{01} + p_{01}q_{01})^{\#y_{01}} \\
& (p_{10}q_{00} + p_{00}q_{10} + p_{10}q_{10})^{\#y_{10}} \cdot \\
& (p_{11}q_{00} + p_{11}q_{01} + p_{10}q_{01} + p_{11} \\
& q_{10} + p_{01}q_{10} + p_{11}q_{11} + p_{10}q_{11} + p_{01}q_{11} + p_{00}q_{11})^{\#y_{11}},
\end{aligned}
$$

where $\#y_{ij}$ is the number of observations $(i,j), i,j = 0, +$ in Group A and $\#\varepsilon_{ij}$ is the number of observations $(i,j), i,j = 0, +$ in Group B. By the MLE, the estimates and the corresponding 95% asymptotic confidence intervals for $p$s and $q$s are as shown in **Supplementary file 1e** and **Supplementary file 1f**.

Thus, the estimated distribution of the possible four types of zebrafish corresponding to Group A-1 to Group A-4 is shown in **Appendix 5—table 3**.

**Appendix 5—table 3.** The estimated probability distribution of each lineage.

| Type | 1 | 2 | 3 | 4 |
|---|---|---|---|---|
| Prob. | 0.4379 | 0.0000 | 0.2841 | 0.2780 |

The results show that the estimates of the probability for Type 1–4 zebrafish are 0.4379, 0, 0.2841 and 0.2780, respectively. It indicates that from the 95% asymptotic confidence intervals, we have 95% confidence to say that Type 1, 3 and 4 zebrafish really exists, while because 0 is in the confidence interval of $p_{10}$, we do not have 95% confidence to say that Type two zebrafish exists. According to the lineages associated with each type of zebrafish, these results suggest that there are at least two different classes of non-HSC hematopoietic progenitors which have distinct lineages. One type of progenitors can differentiate into both T lymphocytes and myeloid cells, and the other type would differentiate into myeloid cells only. The T lymphoid unipotent progenitors were not observed.

## Estimated mean of cell numbers in each zebrafish for each type

Based on the aforementioned probability distributions, we can also calculate the estimated mean of GFP$^+$ T lymphocyte and myeloid cell numbers in each zebrafish for each type. First, we estimated the conditional probability of a zebrafish belonging to a certain type conditioning on whether $Y = 0$ or $Y>0$. Then the estimated mean of the numbers of T lymphocytes and myeloid cells in each zebrafish of type $j$ $(j = 1, 2, 3, 4)$ are

$$\frac{1}{\sum_{i=1}^{n} p_j i} \sum_{i=1}^{n} y_{T_i} p_j i \ \ and \ \ \frac{1}{\sum_{i=1}^{n} p_j i} \sum_{i=1}^{n} y_{M_i} p_j i,$$

where $n$ is the zebrafish number in Group A and $p_{ji} = \hat{P}$(the ith zebrafish belongs to Type j $| Y_T = y_T i, Y_M = y_M i)$. The conditional probability can be calculated based on the probability distribution as in **Appendix 5—table 4**.

**Appendix 5—table 4.** The estimated probability distribution of $X, \varepsilon$ and $Y$.

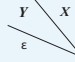

| $Y \backslash X$ $\varepsilon$ | (+,+) | (+,0) | (0,+) | (0,0) |
|---|---|---|---|---|
| (+,+) | 0.0321 | 0.0000 | 0.0208 | 0.0204 |
| (+,0) | 0.0038 | 0.0000 | 0.0024 | 0.0024 |
| (0,+) | 0.1727 | 0.0000 | 0.1121 | 0.1097 |
| (0,0) | 0.2293 | 0.0000 | 0.1487 | 0.1456 |

As calculated, the estimated numbers of GFP$^+$ T lymphocytes and myeloid cells in each Type 1 zebrafish are 9.46 and 21.31, respectively. The estimated number of GFP$^+$ myeloid cells in each Type 3 zebrafish is 6.98. This indicates that a single non-HSC progenitor with lymphoid and myeloid potential can differentiate to about 9 T lymphocytes and 21 myeloid cells in a seven dpf zebrafish, while a single progenitor with myeloid potential only can produce about seven myeloid cells at seven dpf stage.

