## [Decision Letter]

**Acceptance summary:**

To understand the extent and role of cell fate heterogeneity in development and homeostasis, a highly desired goal has been to be able to follow the short- and long-term fate of single cells and their progeny, depending on their precise location and cell type identity within the developing organism at any given time. To this aim, various methods have been developed that used the optical transparency of model organisms such as zebrafish or *C. elegans* to trigger a genetic recombination leading to a switch in fluorescent reporter gene expression via a laser-mediated optical stimulation of the cells of interest. However, none of these methods was yet able to warrant truly single-cell labeling at high efficiency without compromising cell viability. This critical step has now been achieved by the authors of the present paper, via key improvements of a previous method using infrared laser mediated controlled cell heating. They have applied their improved single-cell labelling method with high success to three different cell types in the developing zebrafish, so it will likely be widely applicable to any tissue of transparent animal models.

**Decision letter after peer review:**

Thank you for submitting your article "in vivo single-cell lineage tracing in zebrafish using high-resolution infrared laser-mediated gene induction microscopy" for consideration by *eLife*. Your article has been reviewed by two peer reviewers, including Philippe Herbomel as the Reviewing Editor and Reviewer #1, and the evaluation has been overseen by Didier Stainier as the Senior Editor.

The reviewers have discussed the reviews with one another and the Reviewing Editor has drafted this decision to help you prepare a revised submission.

Summary:

He et al. present a significant improvement of the IR-LEGO method originally introduced by the Kamei lab in 2006 to induce heat shock promoter driven gene expression in specific cells of a live organism by briefly illuminating them with an infrared laser. They manage to obtain a high efficiency of single-cell labeling well above the spontaneous activation of the HS promoter, while preserving cell viability – which was an issue of this method. The optimal conditions of illumination are fine-tuned by using a new method of real-time, high spatial resolution thermometry based on the fluorescence ratio of two fluorophores, one of which is temperature sensitive.

Essential revisions:

1) The narrow margin between labelling efficiency and cell damage was a major limitation in the various studies that used the original IR-LEGO method in various tissues and organisms. Therefore it should be added to the three other "fundamental challenges" listed in the Introduction.

Then in Figure 2—figure supplement 2, we learn that the authors' own set-up, if applied as single-point heating, kills a muscle cell in 2 sec. of illumination! Should we understand that in their previous papers using similar wavelength and power, but 2 min of presumably single-point illumination (Xu, 2015, Tian, 2017, He, 2018) the rate of cell death was actually very high, even though this was not mentioned? In the present work, the authors should devote more than just one sentence to this essential point in the Results section. It deserves at least a whole paragraph, in which they will notably document e.g. the frequency of cell death/damage following point vs. 8 x 8 µm scanning illumination in the various tissues examined, including for the HE cells which are the main target of this and their previous work (Tian, 2017). Also, how was such a long illumination chosen – i.e. how did it influence labeling efficiency vs. cell viability?

Then for the Discussion section, a question remains: molecules diffuse rapidly within the scanned targeted cell, so how can the 8 x 8 µm scanning make such a difference – i.e. less damage to the cell? The authors should discuss this and propose explanations.

2) The simulations of heat diffusion in Appendix 2 and Figure 1—figure supplement 1 are for a 2D diffusion field. But at steady state the diffusion equation reads *∇*^2^*T* = 0; a 3D solution does exist: δT(r) ~1/r (as fohe potential away from a point charge). It would be interesting to compare that solution to the data.

3) The temperature gradient obtained in the z axis is less steep than in the xy plane, implying that more than one cell will often be labeled along the z axis (unless the targeted cells belong to a structure like the hemogenic endothelium studied here, which is basically 1 cell thick). Thus, in Figure 2—figure supplement 4B, the histograms should also be done in the z direction, and we suggest that for a dense tissue such as the muscle, the authors indicate whether the 45% of non-single-cell labeling is due to additional labeling in the z axis vs. x/y plane.

On the same issue: using a 2-photon laser excitation at 670 or 740 nm might yield a large absorption at the focal point and a much more local heating in the z axis. Why not choose that strategy? This could be discussed.

4) A steep temperature gradient is obtained with 3% agarose, and even more so in zebrafish tissue, but none in water. The authors invoke the higher thermal conductivity of water, but the latter is actually only slightly higher than that measured in the literature for 3% agarose or live tissues (0.6 vs. ~ 0.5). (Similarly, I guess their simulation of Appendix 2 would have also produced a gradient with a thermal conductivity of 0.6 instead of 0.5.…?) So some other factor must be invoked: e.g. convection occurring in water?

5) In the Discussion (3rd paragraph), the authors write "However the efficiency of single HE labeling is relatively low (29.3%)". But in the referred Supplementary file 1B and Figure 2—figure supplement 4C, that is not what we see: it is 55.6% – hence higher than for the myocytes! Which value is correct? (If it is the latter, then the rest of the paragraph becomes irrelevant).

---

## [Author Response]

Essential revisions:1) The narrow margin between labelling efficiency and cell damage was a major limitation in the various studies that used the original IR-LEGO method in various tissues and organisms. Therefore it should be added to the three other "fundamental challenges" listed in the Introduction.

We thank the reviewer’s advice. Indeed, the potential induction of cell damage is a crucial issue for IR laser heat shock. We have included this point as one of the fundamental challenges of the IR-LEGO technique in the Introduction of revised manuscript.

Then in Figure 2—figure supplement 2, we learn that the authors' own set-up, if applied as single-point heating, kills a muscle cell in 2 sec. of illumination! Should we understand that in their previous papers using similar wavelength and power, but 2 min of presumably single-point illumination (Xu, 2015, Tian, 2017, He, 2018) the rate of cell death was actually very high, even though this was not mentioned? In the present work, the authors should devote more than just one sentence to this essential point in the Results section.

As pointed out by the reviewer, in our previous study, we applied a relatively long-time (2 mins) illumination of IR laser for efficient labeling of bulk cells in a large region. However, cell damage was found with much shorter illumination time (3s in muscle) using our current IR-LEGO system designed for single-cell labeling. The distinct effect on cell damage is largely due to the focusing capability of objective lenses used in two IR-LEGO systems, respectively. The previous system loosely focuses the IR laser into a large region for bulk cell labeling, while the new IR-LEGO system reported in this work tightly focuses the laser into an extremely small region for single-cell labeling. In general, the focal spot area of a focused laser can be calculated as follows (Pawley 2006):

Afocal=πw2,w≈λπNA

where λ is the wavelength of IR laser, NA is the numerical aperture of objective lens, and w is the waist of IR laser beam (the radius of the circular cross-section of Gaussian beam at the focal plane). Therefore, the numerical aperture (NA) of an objective lens determines its light focusing/gathering capability and defines the focal spot size of a laser beam. Large-NA lens produces a tightly focused laser beam with small focal spot size and high power density, and vice versa. In current IR-LEGO system for single-cell labeling, we use a high-NA (1.15) objective to tightly focus the IR laser into an extremely small spot (~ 0.4 µm^2^) inside cell and generate highly localized heating to build a high gradient of temperature inside cell. In contrast, the NA of the IR-LEGO system used in previous studies (*Xu, 2015, Tian, 2017, He, 2018*) is about 0.15, and the corresponding focal spot area is as large as 26 µm^2^. The power density at laser focal point of new system is estimated about 60 times greater than the old IR-LEGO system, leading to cell damage over much shorter time of IR laser illumination. On the other hand, the loosely focused IR laser in the old IR-LEGO system is more unlikely to cause cell damage, making the 2 min irradiation endurable and efficient for the bulk labeling of HEs. In this work, we applied 32s heating by quickly scanning the IR laser beam over an area of 8 µm × 8 µm, equivalent to 256 × 256 pixels, to avoid cell damage (please see further discussion below). This issue has been addressed in the revised manuscript (Discussion, first paragraph).

It deserves at least a whole paragraph, in which they will notably document e.g. the frequency of cell death/damage following point vs. 8 x 8 µm scanning illumination in the various tissues examined, including for the HE cells which are the main target of this and their previous work (Tian, 2017). Also, how was such a long illumination chosen – i.e. how did it influence labeling efficiency vs. cell viability?

To address the reviewer’s comment, we designed a series of experiments (subsection “Zebrafish Preparation and in vivo Heat Shock”, last paragraph) to study the cell damage by new single-cell labeling system under different heat-shock conditions, as well as by the previous IR-LEGO system under 2min IR laser illumination. We found that neither the scan-heating method in new system nor old method caused obvious damage in targeted HE cells in comparison with the control groups without heat-shock treatment. However, the single point heat-shock without scanning IR laser by the new system caused significantly higher cell death ratio. We presented these results in Figure 4—figure supplement 1) and described them in the Discussion (first paragraph).

We chose the heat shock duration time based on a balance between labeling efficiency and cell damage. In our previous work using low power density heat shock with 2-min illumination, we found that it produced high efficiency for bulk labeling of HE cells without causing obvious cell death. For our new IR-LEGO system with high-NA objective, we scanned the laser focal spot constantly over an area of 8 µm × 8 µm in a cell. This scan-heating mode avoids the quick heat accumulation at single point and effectively reduces the cell damage. We found that scan-heating over 32s produced acceptable efficiency of HE labeling without causing cell damage.

Then for the Discussion section, a question remains: molecules diffuse rapidly within the scanned targeted cell, so how can the 8 x 8 µm scanning make such a difference – i.e. less damage to the cell? The authors should discuss this and propose explanations.

Cell is a mixture of water, large molecules, such as proteins, and organelles. Its thermal conductivity and molecule mobility/convection is lower than water, meaning that heat can be accumulated quickly. This is why a tightly focused IR laser can build up a high gradient of temperature inside cell and achieve single cell labeling. However, quick accumulation of heat at single point also could increase local temperature rapidly and cause cell damage as shown in our experimental results. As suggested by the reviewer, we have discussed this issue in the revised manuscript (Discussion, first paragraph).

*2) The simulations of heat diffusion in Appendix 2 and Figure 1—figure supplement 1 are for a 2D diffusion field. But at steady state the diffusion equation reads ∇*^2^*T = 0; a 3D solution does exist: δT(r) ~1/r (as for the potential away from a point charge). It would be interesting to compare that solution to the data.*

As suggested by the reviewer, we used a 3D heat diffusion model for the simulation of temperature distribution. The laser irradiation is absorbed and characterized by an internal heat generation term (q'). Thus the heat diffusion equation for the medium is:

∇2T+q'k=1α∂T∂t,

where T is the temperature, K is the thermal conductivity of the medium, q' is the intensity of thermal generator, α is the thermal diffusivity of the medium. In the case of steady-state (∂T∂t0) and uniform radial conduction, the heat equation can be represented in spherical coordinates:

1r2∂∂rkr2∂T∂r+q'=0,

where r is the distance from the point thermal generator. As indicated by the reviewer, this differential equation has one particular solution, T1=c/r (c is a constant). Here we specified the temperature boundary condition as Tr=200μm=25℃, and the corresponding temperature distribution along radial direction was plotted as in Author response image 1 (the blue curve).

Compared to the result calculated by the finite-difference method (the red curve; same parameters of thermal generator and medium), this 3D model exhibits a lower peak temperature with sharper gradient. A possible reason for the difference between two models could be their boundary conditions. Specifically, the 3D model has a spherical temperature boundary, while the finite-difference 2D model has a line boundary at 100 µm away from the point thermal generator (Figure 1—figure supplement 1). In fact, the line boundary is closer to the experimental scenario in which the cover glass below the laser focal point serves as a conduction surface. We believe that it would be more accurate to use the finite-difference method for the simulation model.

3) The temperature gradient obtained in the z axis is less steep than in the xy plane, implying that more than one cell will often be labeled along the z axis (unless the targetedcells belong to a structure like the hemogenic endothelium studied here, which is basically 1 cell thick). Thus, in Figure 2—figure supplement 4, the histograms shown in panel B should also be done in the z direction, and we suggest that for a dense tissue such as the muscle, the authors indicate whether the 45% of non-single-cell labeling is due to additional labeling in the z axis vs. x/y plane.

We thank reviewers for the suggestions. In the single-cell labeling demonstrations, it is most challenging to achieve single-cell labeling in muscle (54.5% efficiency) since muscle has the highest cell density. In contrast, other tissues would not suffer from the axial temperature due to sparse cell distribution (neurons with 100% efficiency) or small tissue thickness (*coro1a*^+^ leukocytes with 77.8% efficiency). Therefore, we measured the temperature distribution in z direction during myocyte heat shock, as shown in Figure 2—figure supplement 4C. As can be seen, the average temperature at 10 µm away from the IR laser heating center along z axis is about 37℃, close to the temperature for effective cell labeling. As a consequence, more than one myocyte along z axis were labeled in some zebrafish (Figure 2—figure supplement 4E, marked by the axially elongated DsRed signals), implying that the relatively smaller axial temperature gradients will indeed raise the labeling possibility of cells along z axis. Nevertheless, the optimized heat shock condition has successfully confined the thermal energy in a near single-cell dimension and thus achieved single-cell labeling in more than a half myocytes. (Figure 2—figure supplement 4F).

On the same issue: using a 2-photon laser excitation at 670 or 740 nm might yield a large absorption at the focal point and a much more local heating in the z axis. Why not choose that strategy? This could be discussed.

As to the two-photon strategy, in principle, it could be achieved by using mid-infrared (MIR) lasers (i.e. 2684nm or 2960nm) of which the wavelengths are about two times of the current wavelengths. However, the absorption coefficients of water at the MIR region are extremely high (more than 100 times higher than that at 1342nm or 1480nm), so almost all the MIR light will be absorbed by the water in the agarose gel (which is used to mount zebrafish) and surface tissues of zebrafish, making it impossible to penetrate deep through tissues (the balance between absorption coefficient and penetration depth was discussed in Appendix 1). Therefore, we adopted the near-infrared laser (1342nm) heat shock based on linear absorption of water in tissues.

4) A steep temperature gradient is obtained with 3% agarose, and even more so in zebrafish tissue, but none in water. The authors invoke the higher thermal conductivity of water, but the latter is actually only slightly higher than that measured in the literature for 3% agarose or live tissues (0.6 vs. ~ 0.5). (Similarly, I guess their simulation of Appendix 2 would have also produced a gradient with a thermal conductivity of 0.6 instead of 0.5.…?) So some other factor must be invoked: e.g. convection occurring in water?

We thank the reviewer to point out the problem. Indeed, the conductivity coefficients of agarose and tissues are only slightly larger than that of water, and the simulation result also suggests that the difference in conductivity alone cannot result in the dramatically different temperature distributions measured by the fluorescent thermometry. We agree that in addition to thermal conductivity, the convective heat transfer should also play a crucial role in the small temperature gradient in water. We clarified this in the revised manuscript (subsection “Single-cell Labeling in Zebrafish”, first paragraph).

5) In the Discussion (3rd paragraph), the authors write "However the efficiency of single HE labeling is relatively low (29.3%)". But in the referred Supplementary file 1B and Figure 2—figure supplement 4C, that is not what we see: it is 55.6% – hence higher than for the myocytes! Which value is correct? (If it is the latter, then the rest of the paragraph becomes irrelevant).

We thank the good question from reviewers. Actually, the data shown in Figure 2—figure supplement 4 and Supplementary file 1B have no connection with the HE lineage tracing assay. In the ‘Single-cell Labeling in Zebrafish’ part of Results, to test the single-cell labeling efficiency in blood cells, we directly labeled *coro1a*^+^ leukocytes at the AGM or PBI (referred to as ‘hematopoietic cells’ and ‘vessel’ in the original manuscript), which have 55.6% labeling efficiency (results shown in Figure 2H, Figure 2—figure supplement 4 and Supplementary file 1B). This is independent and distinct from the HE lineage tracing assay presented in the ‘Tracing Single HE and Progenies’ part, in which the single *kdrl*^+^ HEs at the PBI were labeled with 29.3% efficiency. To avoid confusion, in the ‘Single-cell Labeling in Zebrafish’ part, we referred to the *coro1a*^+^ cells as ‘leukocytes’. The corresponding modifications have been made in the revised manuscript (subsection “Single-cell Labeling in Zebrafish”, last paragraph; Figure 2 legend; Discussion, third paragraph; Figure 2—figure supplement 4; Supplementary file 1B).

References:

1) Pawley, J. Handbook of Biological Confocal Microscopy. (Springer Science & Business Media, 2006).